# Mechanical impact of epithelial—mesenchymal transition on epithelial morphogenesis in *Drosophila*

Mélanie Gracia [1], Sophie Theis [1,2], Amsha Proag[1], Guillaume Gay[2], Corinne Benassayag[1] & Magali Suzanne [1]

Epithelial—mesenchymal transition (EMT) is an essential process both in physiological and pathological contexts. Intriguingly, EMT is often associated with tissue invagination during development; however, the impact of EMT on tissue remodeling remain unexplored. Here, we show that at the initiation of the EMT process, cells produce an apico-basal force, orthogonal to the surface of the epithelium, that constitutes an important driving force for tissue invagination in *Drosophila*. When EMT is ectopically induced, cells starting their delamination generate an orthogonal force and induce ectopic folding. Similarly, during mesoderm invagination, cells undergoing EMT generate an apico-basal force through the formation of apico-basal structures of myosin II. Using both laser microdissection and in silico physical modelling, we show that mesoderm invagination does not proceed if apico-basal forces are impaired, indicating that they constitute driving forces in the folding process. Altogether, these data reveal the mechanical impact of EMT on morphogenesis.

[1] LBCMCP, Centre de Biologie Intégrative (CBI), Université de Toulouse, CNRS, UPS, Toulouse 31062, France. [2] Morphogénie Logiciels, 32110 St Martin d'Armagnac, France. Correspondence and requests for materials should be addressed to G.G. (email: guillaume@damcb.com) or to C.B. (email: corinne.ben-assayag@univ-tlse3.fr) or to M.S. (email: magali.suzanne@univ-tlse3.fr)

Epithelial−mesenchymal transition (EMT) is an evolutionary conserved cellular process occurring in multiple occasions during development, from early embryogenesis, where it plays a fundamental role during gastrulation in the formation of new layers, to the delamination of neural crest, the formation of somites or the development of the heart. EMT includes the progressive loss of epithelial characteristics, together with the gain of mesenchymal markers. In addition to its important functions during development, EMT is also involved in pathological contexts such as metastasis dissemination[1,2]. A well-known EMT inducer is the gene *snail*[3], very conserved during evolution and first described in *Drosophila*[4]. Snail is now forming a large family of genes in metazoans, involved both in developmental and pathological EMT.

Strikingly, during gastrulation, EMT is often associated with tissue invagination. It is the case for example at the primitive streak in amniotes, at the blastopore lip in *Xenopus* or at the ventral furrow in *Drosophila*. However, the nonautonomous influence of cell ingression on tissue bending has never been directly addressed.

To test the potential impact of EMT on the surrounding epithelium, we first characterize the cellular dynamics associated with EMT. We focus on the initiation of the EMT process, which corresponds to the very beginning of cell delamination. To do so, we induce EMT ectopically in a naive tissue (leg imaginal disc) by ectopic Snail expression. In this context, we observe that prior to delamination, each cell maintains strong cell−cell adhesion and generates a force orthogonal to the apical surface leading to the deformation of the epithelium around the collapsing apex. We further find that inducing EMT ectopically is sufficient to induce tissue remodeling in a naive tissue, as shown by the formation of ectopic invaginations. We then ask if this is a general feature of EMT and decide to look at an endogenous EMT process. We turn to mesoderm invagination in the embryo, a well-characterized EMT process dependent on the early expression of Snail in mesoderm cells. We find that when mesoderm cells constrict their apex and start their delamination, they form a particular apico-basal cable-like structure of myosin II and generate an inward force. We further test the importance of this force in mesoderm invagination and discover that it constitutes one of the main driving forces required for this remodeling process. Finally, we test the relative importance of apical constriction versus apico-basal traction force in tissue invagination in a 3D vertex model. We find that even very strong values of apical force are not sufficient in this theoretical model to induce tissue folding, whereas apico-basal traction is crucial to drive invagination.

Altogether, this work reveals that cells undergoing EMT are not passively expulsed from the epithelium. Rather, before their delamination, they generate forces orthogonal to the plane of the epithelium, and thus actively participate in tissue folding.

## Results

### EMT induces tissue folding

In order to study EMT dynamics, we first look for a way to induce EMT ectopically. We show that overexpressing the EMT inducer Snail in a group of naive cells from leg imaginal discs (either in clones or in the *apterous* domain) was sufficient to recapitulate the hallmarks of EMT including the extrusion from the epithelial sheet or delamination (Fig. 1a), the progressive loss of cell−cell adhesion (Fig. 1b and Supplementary Fig. 1a–c) and the acquisition of migratory properties (Fig. 1c). We then characterized their individual dynamics in the monolayer epithelium forming the leg imaginal discs. Snail-expressing cells initially conserve strong adhesion with their neighbors (Fig. 1a) and constrict apically before progressively leaving the epithelial sheet (Figs. 1a and 2a).

Interestingly, during this apical constriction phase, an apico-basal myosin II accumulation forms within Snail-expressing cells (Fig. 2a, white arrowhead). In living samples, we could observe that this apico-basal accumulation of myosin (hereafter named "cable" for simplification) is systematically associated with a local deformation of the apical surface around the constricting cell, which indicates that it produces a traction force at the onset of the delamination process (Fig. 2b, c, Supplementary Movie 1). These deformations appear to be transient, consistent with the final loss of adhesion of the delaminating cells (Fig. 2b and Supplementary Fig. 1a–c) and their final extrusion (Figs. 1a and 2a). We further notice that within a group of Snail-expressing cells, cells do not start constricting their apex synchronously (Fig. 2b, d, apical views). Indeed, a few cells start apical constriction before the other and the apico-basal myosin cable formed specifically in the most advanced constricting cells (Fig. 2b).

At a broader scale, following cellular dynamics in big clones of Snail overexpression, we found that cells located in the vicinity of a cell exerting an apico-basal pulling force, reduce their apical surface more rapidly than the rest of the clone (Fig. 2d, apical views). Indeed, a gradient of decreasing constriction rates is observed from the closest neighbors to the farthest ones (Fig. 2e). This reorganization eventually results in the formation of an invagination around the first constricting cells (Fig. 2d, sagittal views, white arrow, Supplementary Movie 2). Finally, as the invagination of Snail-expressing clones progresses, the surrounding tissue (which does not express Snail) is also deformed, leading to the formation of ectopic folds composed of Snail-expressing and nonexpressing cells (compare Fig. 2g with 2f). This even leads to stable morphogenetic perturbations that are conserved even after the delamination of the whole clone (Supplementary Fig. 1h, i). Consistently, ectopic invagination can also be observed between various small groups of cells expressing Snail (Supplementary Fig. 1d–g, Supplementary Movie 3). This points to an interesting nonautonomous effect of Snail-driven forces, which is indicative of the mechanical influence of Snail-expressing cells on the surrounding tissue.

Altogether, these results show that at the onset of EMT, cells generate myosin II-dependent apico-basal forces while constricting their apex and pull down transiently the apical surface of the epithelium, before delaminating. At the tissue scale, ectopic EMT appears sufficient to drive ectopic folding, suggesting that the force generated by predeliminating cells constitutes a driving force for tissue remodeling.

We then asked if this force, generated orthogonally to the plane of the epithelium, was a general feature of cells undergoing EMT and if it could be involved in endogenous EMT-associated folding. We chose to focus on mesoderm invagination, a well-known morphogenetic process that combines tissue invagination with Snail-induced EMT.

### Mesoderm cells form myosin II apico-basal cables

Mesoderm invagination is the first morphogenetic movement that takes place in the *Drosophila* embryo and leads to the formation of a multilayered structure from an initial monolayer blastoderm. It has been initially described as divided in two phases: the first one is ventral furrow formation and includes the successive steps of apical constriction of Snail-expressing ventral cells, which first leads to a change in epithelium curvature, followed by a V-shaped invagination and the formation of a tube of cells that remain attached to each other (Fig. 3a); the second phase corresponds to the loss of epithelial characteristics of mesoderm cells and to their dispersion as they finally form a monolayer underlying the ectoderm[5]. Thus, based on the appearance of mesenchymal cells, which corresponds to the final stage of EMT, the transition has

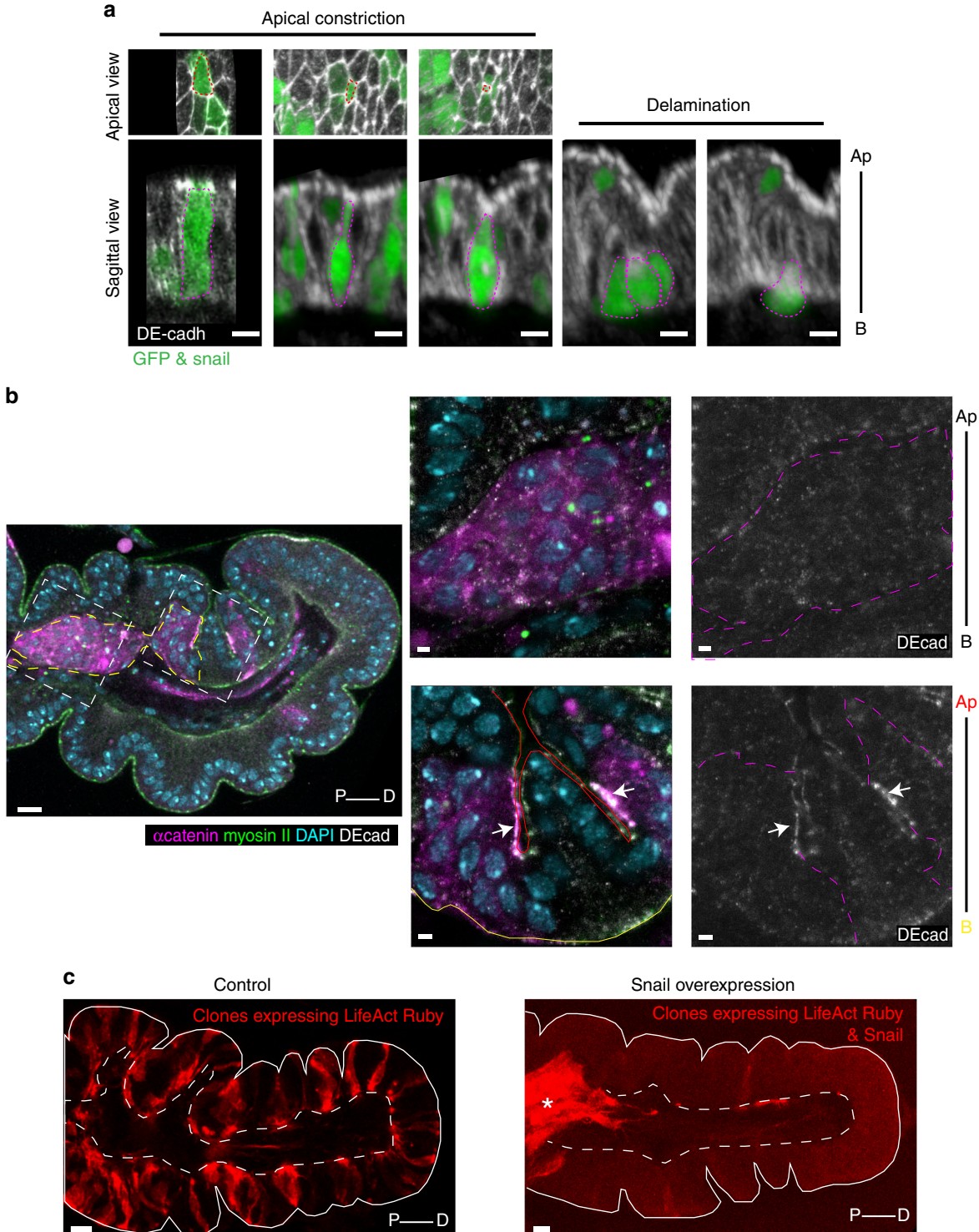

**Fig. 1** The overexpression of Snail is sufficient to induce ectopic EMT. **a** Snail-expressing clones (marked by GFP) generated in leg imaginal discs. Images of fixed tissues (top: apical view, bottom: sagittal view), showing the successive steps of cell apical constriction and delamination. Snail-expressing cells are outlined in dashed magenta and epithelium is marked by DE-cadherin in white. Scale bar: 5 μm. **b** General view and close-up of big clones of cells expressing Snail (marked by the α-catenin in magenta or outline in dashed magenta line) in fixed imaginal leg discs ($n = 19$) showing the loss of DE-cadherin in clones that have delaminate and migrate below the epithelium (see left frame on the general view and top close-up), while clones inserted in the epithelial sheet keep strong DE-cadherin (see right frame in the general view and white arrows in the bottom close-up). DE-cadherin is in white, myosin II in green and nucleus in cyan. Apical and basal side of the epithelium are outlined in red and yellow, respectively. See also Supplementary Fig. 1a–c. Scale bar: 10 μm in the general view and 2 μm in the close-up. **c** Clones of UAS-life-act::Ruby (left, control) or UAS-life-act::Ruby; UAS-Snail (right) highlighting the migratory properties acquired by Snail-expressing cells. Note that the clones (red cells) are well integrated in the monolayer epithelium (outlined in white) in the control (left, UAS-life-act::Ruby), while they have delaminated and migrated below the epithelial sheet when Snail is overexpressed (right, UAS-life-act::Ruby; UAS-Snail; asterisk) as seen in the general view in (**b**). Scale bar: 10 μm. Ap apical, B basal, P proximal, D Distal

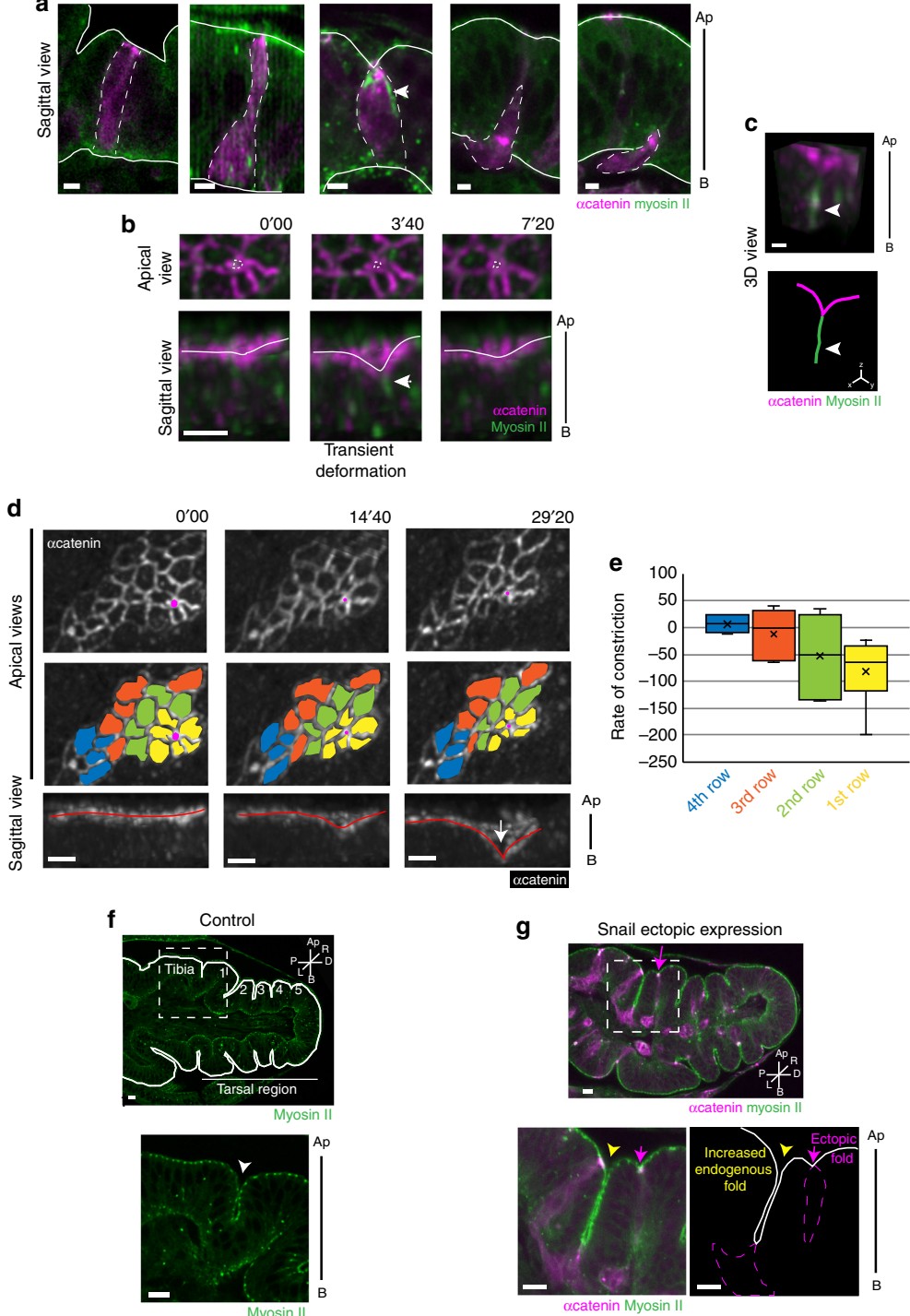

been described as occurring right after mesoderm invagination. However, the initiation of the transition has never been described and since the invagination lasts only a few minutes, it is tempting to speculate that EMT could start during invagination. Consistently, several studies revealed that, although apical constriction is globally strongly coordinated, some cells start to constrict their apex before the others[5–8]. This apical constriction has been shown to coincide with a basal repositioning of the nuclei[9] and strongly suggests that these cells will be the first to delaminate. As a result, the asynchrony of apical constriction in mesoderm cells at the very beginning of the invagination might generate an asynchrony in cell delamination. This leads us to ask if, similarly

to what has been described during primitive streak formation in the chick[10], EMT would start earlier than previously anticipated in mesoderm cells by sporadic delaminations, followed by a massive cell ingression at the end of invagination. Consistent with this hypothesis, some apical surfaces disappear sporadically at an early stage of invagination (Fig. 3b). To further support this point, we asked if some cells from the mesoderm are delaminating before the end of the invagination process. We confirmed the heterogeneity in nuclei positioning along the apico-basal axis of mesoderm cells at the very beginning of the invagination (Fig. 3c, top), as described previously[9]. We further show that although the majority of cells from the mesoderm stay interconnected until the

**Fig. 2** Cells undergoing EMT generate an apico-basal force actively involved in tissue folding. **a–c** Snail-expressing clones (marked by α-catenin-RFP in magenta) generated in sqh::sqh::GFP leg imaginal discs. **a** Images of fixed tissues recapitulating the successive steps of the delamination of cells undergoing EMT. A myosin II apico-basal structure is formed in a constricting cell (white arrowhead). Scale bar: 3 μm. **b** Time-lapse of a Snail-expressing clone, showing asynchronous apical constriction (apical view). While a cell constricts its apex (white dashed outline, apical view), a transient deformation appears at the apical surface (white outline, sagittal view), concomitant to an apico-basal myosin II structure (white arrowhead). Scale bar: 3 μm. **c** 3D reconstruction of a Snail-expressing clone and corresponding scheme showing the apical deformation generated by a pulling cell expressing Snail and the concomitant formation of a myosin II apico-basal structure (white arrowhead). See also Supplementary Movie 1. Scale bar: 1 μm. **d** Time-lapse images of a Snail-expressing clone. First (yellow), second (green), third (orange) and fourth (blue) row of neighbors from the delaminating cell (magenta) are color-coded to highlight their gradual constriction rate (apical views). The whole tissue ultimately produces a stable invagination (red outline, white arrow). See also Supplementary Movie 2. Scale bar: 3 μm. **e** Quantification of apical constriction rate in first (yellow), second (green), third (orange) and fourth (blue) row of neighbors from the delaminating cell (see Methods; $n = 22$ cells). **f** Sagittal sections of a control sqh::sqh::GFP leg disc and higher magnification of the tibia-T1 fold (arrows) as indicated in the general view (dotted square). Normal folds are outlined in white. Scale bar: 10 μm. **g** Sagittal section of a sqhKI [eGFP] leg disc with Snail-expressing clones (marked by α-catenin::RFP expression in magenta) with an ectopic fold formed in the T1 tarsal segment (magenta arrow). Higher magnifications of the tibia-T1 fold region is shown below. Note the deeper tibia-T1 fold (yellow arrowhead), compare to the control (in **f**). On the scheme (right) the Snail-expressing clones are outlined in magenta and the epithelium in white. See also Supplementary Fig. 1d−i and Supplementary Movie 3. Scale bar: 10 μm. Ap apical, B basal, P proximal, D distal, L left, R right

end of the invagination, some cells start their delamination during the invagination as shown by the presence of Snail-positive cells protruding or below the invaginating epithelial layer (Fig. 3c, bottom). In addition, DE-cadherin is not detected in these extruded cells, suggesting that they have lost their epithelial characteristics (Fig. 3d). Thus, at least for a subset of mesodermal cells, EMT appears to start earlier than previously anticipated.

To identify potential new driving forces generated during mesoderm invagination, we focused on myosin II dynamics during mesoderm invagination. Using a novel knock-in construct allowing to follow the whole pool of myosin (sqhKI[eGFP], see Methods), we first observed, as previously described, a strong accumulation in an apical meshwork responsible for apical constriction (Fig. 4a, ventral views)[8]. After this first apical constriction phase, cells from the mesoderm acquire a "wedge-shape" with a reduced apical surface and a wide basal surface, which leads to a characteristic curved shape of the ventral surface of the embryo (hereafter called "curved shape" based on morphological criteria). Interestingly, following this well-described constriction phase, dynamic apico-basal myosin II cable-like linear structures appear in mesoderm cells, perpendicularly to the apical surface. These myosin II structures are reminiscent of the structures observed in Snail-expressing clones and will be called "cables" hereafter. They first appear sporadically in a few cells, then become more abundant and visible all along the antero-posterior axis of the embryo, although never in all mesodermal cells at once (Fig. 4a–c, transversal and longitudinal views, Supplementary Movies 4, 5). Interestingly, we observed that myosin II cables form during the whole invagination process (Fig. 4d), specifically in mesodermal cells that are well advanced in the apical constriction phase (Fig. 4e, Supplementary Movie 6). Quantification of these structures reveals that they form in about half mesodermal cells during the first half of the invagination process (see Methods), strongly suggesting that each mesodermal cell form this structure at a given point during their delamination process.

These transient structures form sequentially in different mesoderm cells, suggesting the existence of heterogeneity in the mesoderm territory. Since cables are preferentially formed in cells with strongly constricted apex, it is thus tempting to speculate that they are formed specifically in cells well advanced in the apical constriction process that will be the first to delaminate.

**An apico-basal force drive mesoderm invagination**. Since myosin II cables appear specifically during invagination, starting as soon as the ventral part of the embryo adopts a curved shape, we reasoned that they could generate an apico-basal force potentially involved in mesoderm invagination. We first asked if

these structures were generating a force using laser ablation. We could show that they are indeed under tension (Fig. 5a, c and Supplementary Movie 7), while lateral membrane do not appear to be very tensed at this stage (Fig. 5b, c).

We next wanted to test the potential role of apico-basal forces in mesoderm invagination. To impair specifically apico-basal forces without altering apical constriction, we used laser microdissection. We first set up conditions to specifically target apico-basal myosin cables without affecting the apical pool of myosin II and the subsequent apical constriction. We thus restricted laser ablation to the mid-plane of ventral cells (mid-plane cuts) and avoided the anterior-and posterior-most parts of the embryo where the laser cut would have affected the apical surface due to the curvature of the embryo (see ablation set-up in Supplementary Fig. 2a). After mid-plane cuts in the central region of the embryo, the mesoderm eventually invaginates (Fig. 6c, bottom panel), indicating that laser ablation did not affect tissue development. Interestingly however, we noticed a slight delay in the invagination of the targeted central region compared to the intact anterior- and posterior-most regions (Fig. 6b, c, top panel). This is shown by the slightly curved shape of the invagination front line compared to the straight line observed in the control embryos (see bottom scheme in Fig. 6b, compare Fig. 6b with Fig. 6a, see Supplementary Fig. 2c for quantifications). These results might suggest that, at least in these specific conditions, apico-basal forces in the central region do not play a determining role in ventral invagination. Alternatively, we reasoned that the weakness of the phenotype could be due to the remaining myosin II cables formed in the anterior- and posterior-most parts of the embryo, which might compensate for the lack of apico-basal forces in the central region. In order to test this hypothesis, we decided to isolate the central domain of the invaginating mesoderm from the anterior- and posterior-most regions of the embryo (apical cuts, see ablation set-up in Supplementary Fig. 2b). Strikingly, we observed that isolation of the central region per se did not affect mesoderm invagination, showing the robustness of this process (compare Fig. 6e with 6d). We then performed ablation in the mid-plane of the isolated central region of the mesoderm (apical cuts and mid-plane cut). In this context, apical constriction appears mostly unperturbed, although maybe slightly delayed (see quantifications in Supplementary Fig. 2e), as shown by the characteristic pulses of apical myosin II and the curved shape acquired by the mesoderm. However, when mid-plane cuts were performed, invagination was completely impaired in the vast majority of the embryos (compare Fig. 6e–f, with and without mid-plane cuts and see quantifications in Supplementary Fig. 2d, Supplementary Movie 8).

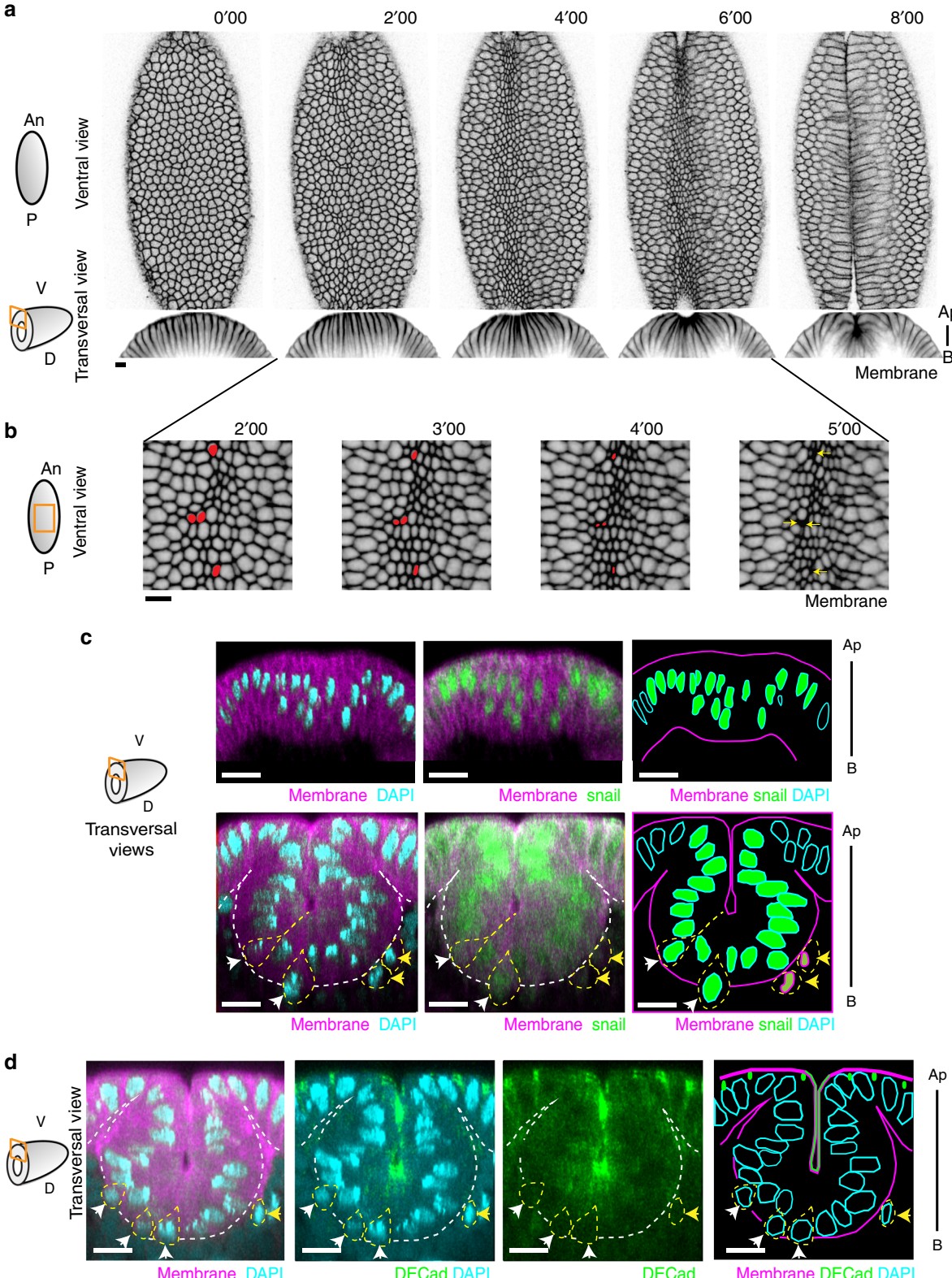

Altogether, these experiments reveal (1) that the ablation of apico-basal structures in the central part of the embryo leads to a delay of invagination, which indicates that preventing apico-basal forces perturbs the invagination process; (2) that the mechanical isolation of the central region by anterior and posterior cuts does not perturb its invagination, which means that invagination can proceed normally without the most anterior and posterior regions; (3) that the mechanical isolation coupled to the ablation of apico-basal structures totally prevents invagination, indicating that in the isolated region, which is perfectly able to invaginate normally in the presence of apico-basal forces, apical myosin is not sufficient to drive invagination and (4) that mid-plane ablation is totally deleterious. Altogether, these experiments reveal the crucial role of apico-basal forces in mesoderm invagination.

**Fig. 3** Cell delamination starts during mesoderm invagination. **a** Image extracted from a movie showing ventral and sagittal views of a PH-mCherry embryo during mesoderm invagination. Scale bar: 10 μm. **b** Time-lapse images of PH-mCherry embryo (ventral view, see scheme) showing strong apical reduction, then apical surface  disappearance (yellow arrows) of cells colored in red during mesoderm invagination. Scale bar: 10 μm. **c** Transversal views of PH-mCherry fixed embryos at early (top) or late (bottom) stages of invagination stained with anti-Snail antibody (n = 17). Note the basal relocation of some nuclei within the mesoderm domain (Snail-positive cells, in green) at early stage and the presence of cells at different stage of delamination (outline in dashed yellow) at later stage: arrowheads point at protruding (white arrowheads) and extruded Snail-expressing cells (yellow arrowheads). Note that Snail expression is reduced in extruded cells. Scale bar: 10 μm. **d** Transversal view of PH-mCherry fixed embryos at late stages of invagination stained with anti-DE-cadherin antibody (n = 11). Note the presence of protruding (white arrowheads) and extruded (yellow arrowheads) cells (outline in dashed yellow). DE-cadherin is not detected in extruded cells. Scale bar: 10 μm. An anterior, P posterior, V ventral, D dorsal, Ap apical, B basal

Since cable ablation prevented folding only after mechanical isolation, we then asked whether apico-basal forces could be transmitted throughout the tissue via the apical surface. To answer this question, we performed a similar set of experiments in which the apical surface was cut only on one side (i.e. posteriorly). In the absence of mid-plane cut, the invagination front formed a straight line, suggesting that invagination rate was rather homogeneous along the antero-posterior axis of the embryo (Supplementary Fig. 3a). However, the invagination was clearly asymmetric when apico-basal forces were impaired in the central region. Indeed, in this case, the nonablated anterior region appeared to drag the ablated central one, which showed a gradient of invagination from anterior to posterior (Supplementary Fig. 3b). These experiments reveal that the apical surface enables long-range force transmission of apico-basal forces and that the absence of apico-basal forces in a particular region can be compensated by the production of apico-basal forces in the neighboring one.

Altogether, these experiments confirm that apical constriction, although responsible for the "curved shape" observed at the onset of invagination, is insufficient to drive the deep V-shaped mesoderm invagination. Furthermore, they reveal that an apico-basal force generated by the mesoderm is essential for this morphogenetic process.

**Apico-basal traction is required for mesoderm invagination in a 3D biophysical model.** To test specifically the contribution of these apico-basal forces in tissue folding, we developed a physical model of the embryo apical junction network based on the vertex model we implemented recently[11]. This 3D model of mesoderm invagination mimics the cellular dynamics observed in the embryo (anisotropic constriction, gradual and asynchronous apical constriction, and apical force propagation). It relies on the minimization of an energy function with three mechanical components: a cell area elasticity, a quadratic contractility term that depends on the cell perimeter and an apico-basal line tension that mimics apico-basal forces (see Methods).

To reproduce the shape of the embryo, the virtual tissue is an ellipsoid of 6000 cells that includes an oval domain of about 850 cells representing the mesoderm (Fig. 7a). In this model, mesodermal cells progressively constrict their apical surface preparing their future delamination (Fig. 7b). Based on our observations of the embryo, we programmed apical constriction to occur (1) gradually from the central-most part of the mesoderm to the more lateral regions (Supplementary Fig. 4b, top), thus reproducing the gradient described previously[6] and (2) with a slight asynchrony, thus mimicking the heterogeneity of apical constriction observed in vivo (compare Fig. 7f with Fig. 3b). Finally, yolk incompressibility is represented by a global volume elasticity (Supplementary Fig. 4a) and the vitelline membrane by a rigid envelope. We first set the different parameters so we could reproduce the curved shape of the tissue as well as the cell apical area distribution observed in vivo at this stage (Supplementary Fig. 4b, bottom).

We then tested the impact of an increase of apical contractility to test the influence of apical constriction alone in this numerical model. In these conditions, although the distribution of cell area within the mesoderm territory is similar to that observed in vivo (see Fig. 7d and compare the upper panel of Fig. 7e with Fig. 7c), and the curved shape nicely reproduced (Fig. 7e, transversal sections), the invagination rapidly reaches a maximum depth and is far from reproducing the depth (Fig. 7g) and the V-shape of the ventral furrow observed in the *Drosophila* embryo (Fig. 7h). These results support the idea that apical constriction is not sufficient to induce tissue invagination. Indeed, although the epithelium is properly incurved, forming the characteristic curved shape, no further invagination is observed, even for very strong values of apical contractility.

In a second step, we implemented the model by adding an apico-basal traction force generated orthogonally to the surface of the epithelium in constricting cells (Fig. 8a, red arrow). Here, as previously, we mimic the heterogeneity of apical constriction observed in vivo (compare Fig. 8e with Fig. 3b) and the distribution of apical area fits even better the one observed in the real embryo when the embryo adopts a curved shape (see Fig. 8c and compare the upper panel of Fig. 8d with Fig. 8b). Interestingly, in this context, using amplitudes of apical and apico-basal forces within the same order of magnitude, invagination progresses normally and forms the characteristic V-shape in a robust manner (Fig. 8g). We tested different values of apical contractility and apico-basal forces and observed that the depth of the invagination increases with the strength of apico-basal force (Fig. 8f, g and Supplementary Fig. 4c).

Given the robustness of this model, we decided to simulate laser-ablation experiments and further test the respective importance of these forces in isolated domains of the mesoderm. Strikingly, tissue dynamics was very similar to what was observed in vivo. Mid-plane cuts in the central region were mimicked by the absence of apico-basal traction in the central region of the virtual embryo, while apical cuts were mimicked by a drop in apical contractility and area elasticity (see Methods). Using these parameters, we found that mid-plane cuts in the central region of the embryo lead to a curved front line of invagination (compare Supplementary Fig. 5a–c with Fig. 6a–c and Supplementary Fig. 2c); that a posterior apical cut together with mid-plane cuts lead to an asymmetric invagination front (compare Supplementary Fig. 5d–f with Supplementary Fig. 3a, b); and finally, that isolation of the central region by anterior and posterior apical cuts together with mid-plane cuts totally abolished the invagination, although the invagination does take place normally when apico-basal traction forces are still present in the central isolated region (see Fig. 9a–e and compare transversal sections in Fig. 9a–c with Fig. 6d–f and Supplementary Fig. 2d).

Thus, simulation results showed that apical tension alone without apico-basal force only leads to a curved-shaped mesoderm, independently of the strength of apical contractility. Only in the presence of apico-basal tension does the mesoderm invaginate, forming the V-shape observed in vivo (compare

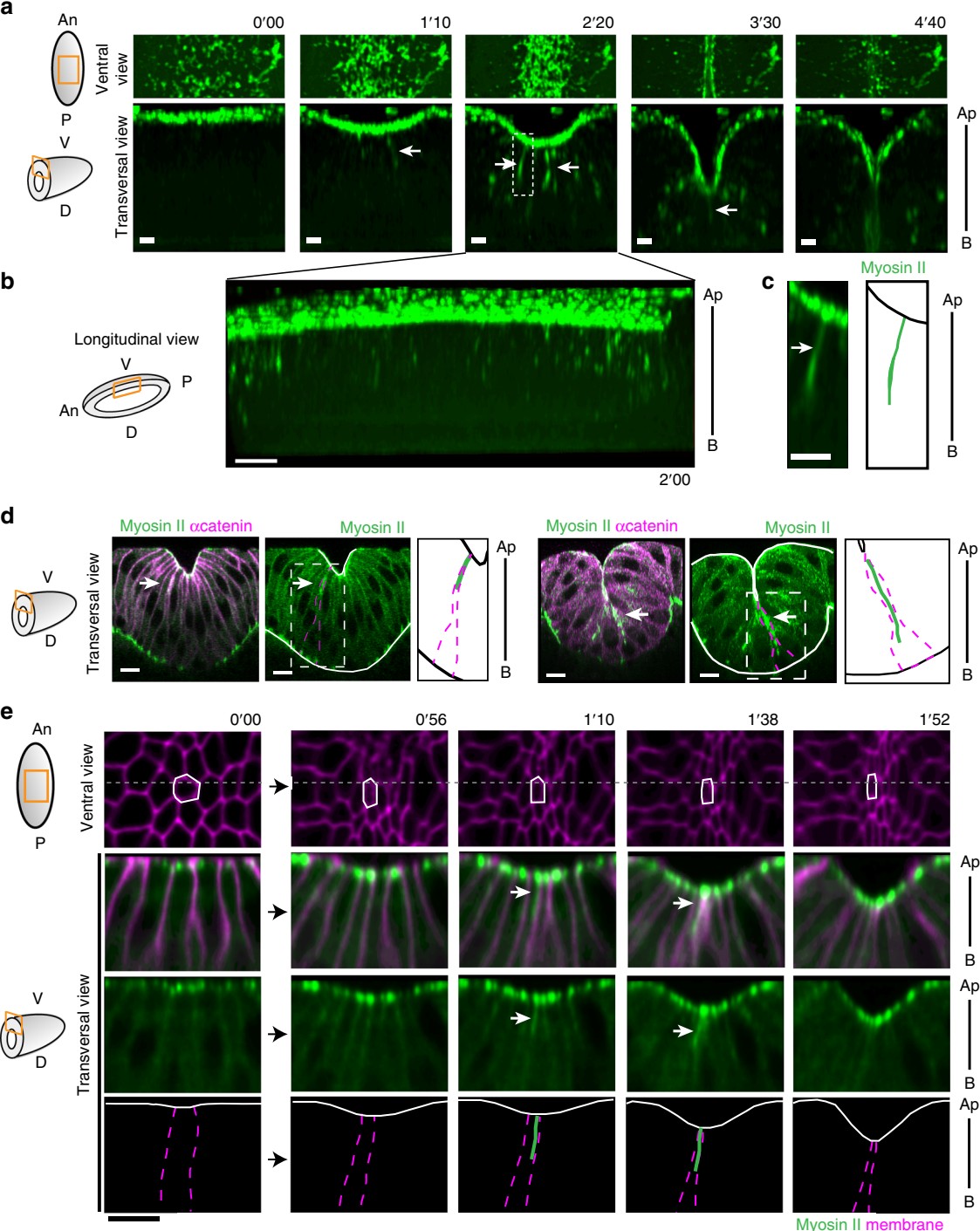

**Fig. 4** Cells from the mesoderm form myosin II apico-basal cables during mesoderm invagination. **a–c** Time-lapse images of the ventral part of a sqhKI [eGFP] embryo during invagination (ventral and transversal views, see schemes, **a**). First, myosin II is accumulated apically (ventral view). Then, transient linear structures perpendicular to the apical surface are formed (white arrows, transversal view), from the onset to the end of invagination (*n* = 82). See Supplementary Movie 4. Scale bar: 10 μm. **b** Longitudinal view of the same tissue at *t* = 2 min showing these apico-basal structure of myosin II all along the embryo. Scale bar: 10 μm. **c** Enlargement of the apico-basal myosin II transient structure (white arrow) in the dashed white framed in (**a**) and the corresponding scheme. See Supplementary Movie 5. Scale bar: 5 μm. **d** Transversal views of the invaginating mesoderm in fixed embryos showing a myosin II cable formed in cells that have just constricted their apex, at early (left) and late (right) stages of invagination (white arrows). The apical and basal surfaces are outlined in white. Right of each panel: Schemes of the framed regions focusing on the apically constricted cell producing the apico-basal structure of myosin II (green line: myosin II cable, magenta dashed line: cell outline, black line: outline of the epithelium). Scale bar: 10 μm. **e** Time-lapse images of mesoderm invagination in a sqhKI[eGFP]; PH-mCherry embryo (ventral and transversal views, see schemes) showing the dynamics of a myosin II cable (white arrow) formed in a cell that constricts its apex (outlined by a white line in the ventral views). Dashed lines indicate the section corresponding to transversal views. Schemes recapitulate myosin II dynamics (bottom panel). See Supplementary Movie 6. Scale bar: 10 μm. An anterior, P posterior, V ventral, D dorsal, Ap apical, B basal

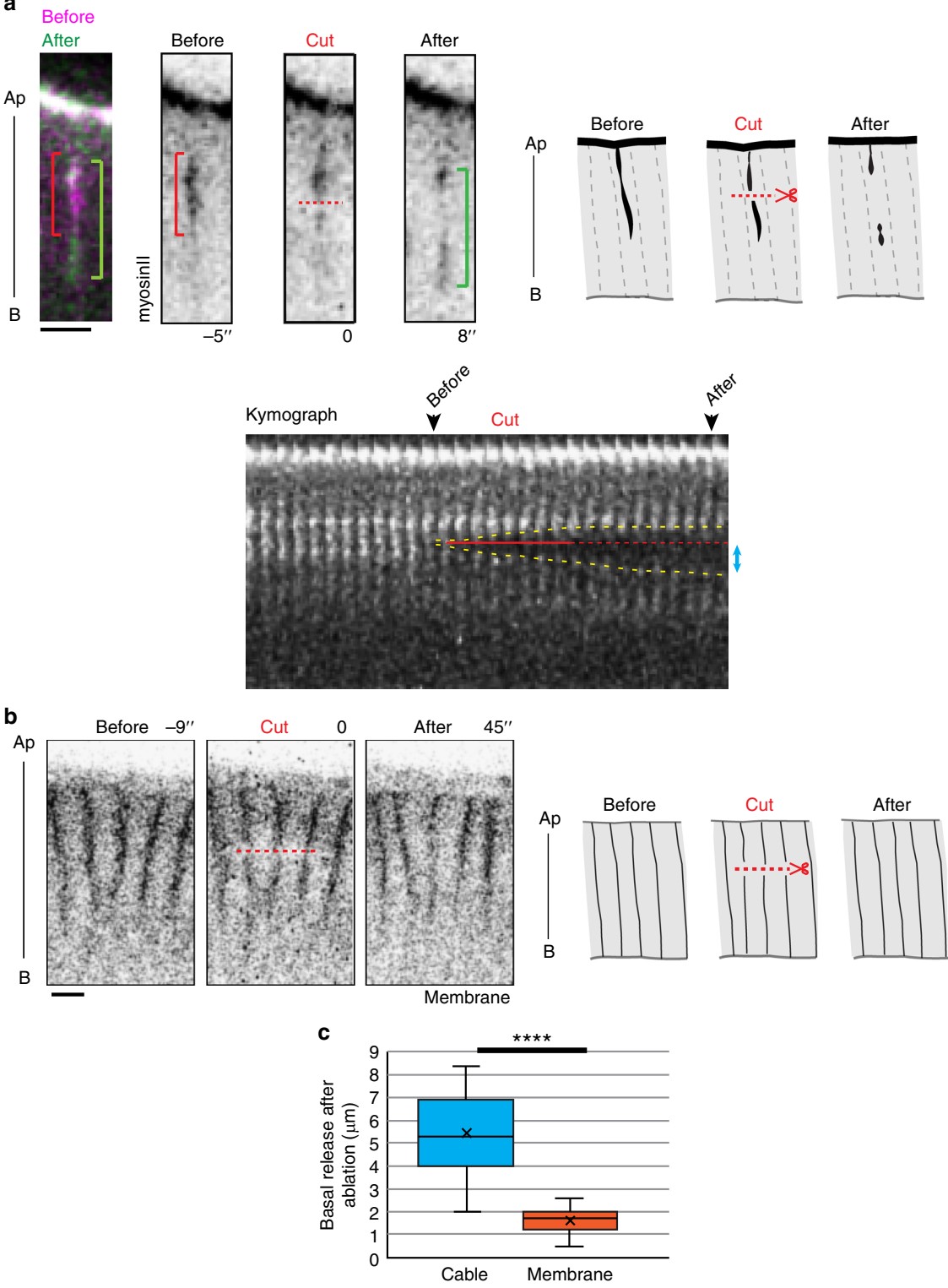

**Fig. 5** Apico-basal cables formed in mesoderm cells are under tension. **a** Laser ablation of apico-basal myosin cable. (top) Images extracted from a movie and corresponding schemes showing transversal views of a KI-sqh::GFP embryo mesoderm cells showing a myosin cable before (red bracket), during and after (green bracket) laser cut (red dotted line). In the kymograph (bottom), the timing of laser cut is indicated by the red line, prolonged by a dotted red line to visualize the site of ablation, and the cable release highlighted by yellow dotted line. Blue arrows indicate the basal release. Scale bar: 5 μm. **b** Control laser ablation of lateral membranes of the embryo (resille::GFP line). Images extracted from a movie and corresponding schemes showing transversal views of the epithelium before, during and after laser ablation. The laser cut is indicated with a red dotted line. Note that the fluorescence at the membranes appear to recover rapidly. Scale bar: 5 μm. **c** Curves of the average recoil observed between the region of ablation and the remaining cable (as shown by the blue arrow in the kymograph) or the remaining membrane. Statistical significance was assessed by the Wilcoxon signed-rank test (control cut (membrane): $n = 11$ embryos; apico-basal cable cut: $n = 11$ embryos; $p < 0.0001$—significant ****). Ap apical, B basal

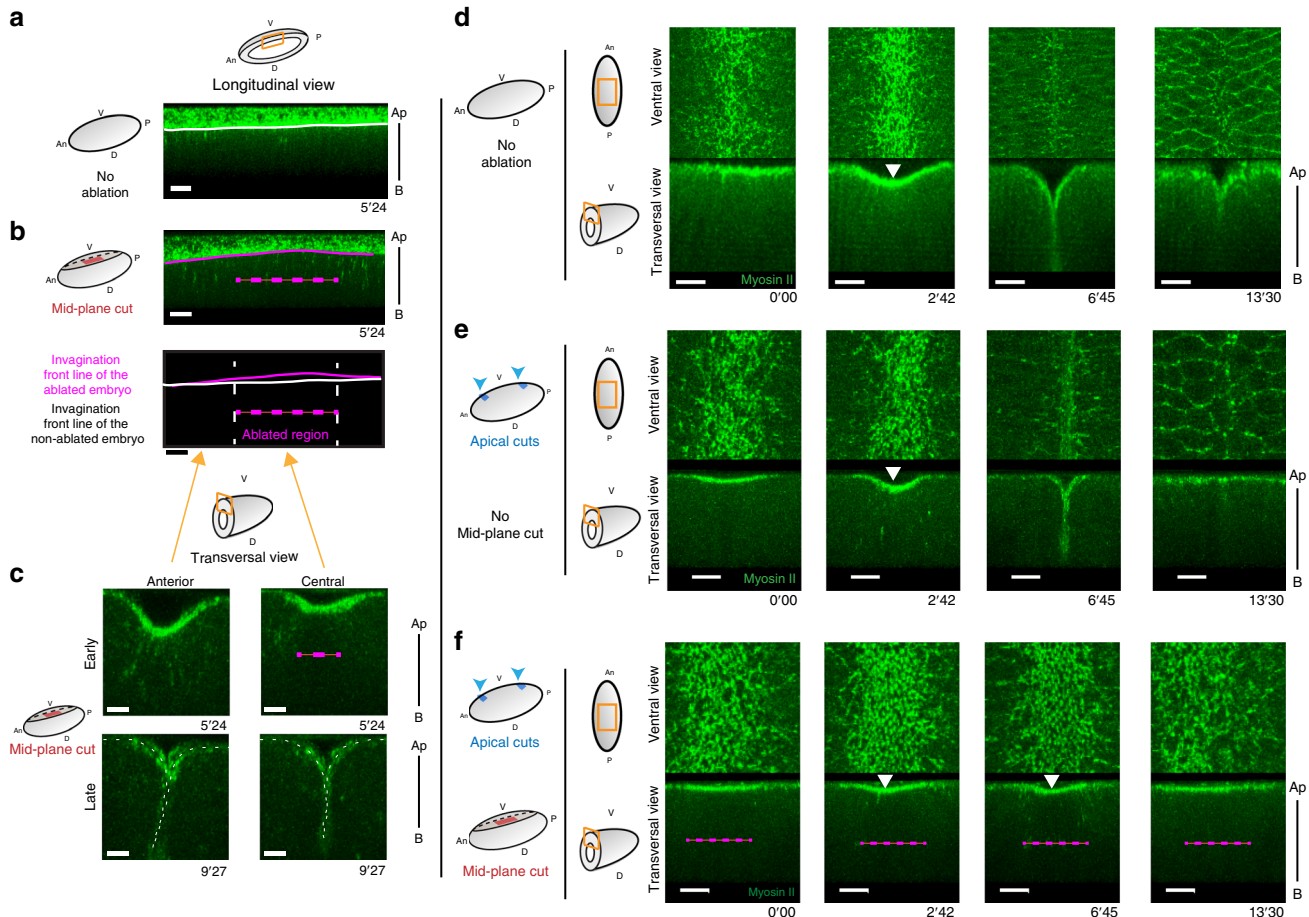

**Fig. 6** An apico-basal force drives mesoderm invagination. Schemes recapitulate the orientation of the embryos and the laser cuts performed in each experiment. **a** Longitudinal views of sqhKI[eGFP] control embryos (*n* = 17), showing the linear shape of the invagination front at early stage. **b** (top) Longitudinal views of sqhKI[eGFP] embryo ablated in the mid-plane region (red rectangle in the scheme and a dashed magenta line in the figure, *n* = 12), showing a slight delay in the central region compared to the anterior one, as shown by the outline of the invagination front in magenta at early stage of invagination. (bottom) Scheme of the invagination fronts in the control (white line, as in **a**) and ablated embryo (magenta line). **c** Anterior (left) and central (right) transversal views of a mid-plane ablated embryo, at two different times of the invagination. Note that invagination proceeds normally as shown by the V-shaped invagination outlined by a white dotted line (bottom), although a delay in the central region can be observed at early stage (top, compare anterior to central region). **d** Ventral and transversal views extracted from a movie of a control sqhKI[eGFP] embryos, showing the different step of mesoderm invagination. **e** The apical surface was cut on the anterior and the posterior sides (blue arrowheads, see left schemes) to isolate the central domain of the mesoderm. Note that the invagination is taking place normally, in term of invagination shape, depth and timing (*n* = 10). **f** After isolation of the central domain (apical cuts as described in **e**), ablation was performed in the mid-plane of the central region of the mesoderm (red rectangle on the scheme, magenta dashed line in the figure). Note that the invagination is totally prevented when apico-basal forces are impaired in the vast majority of the cases (*n* = 11/12). **e**, **f** Note that apical constriction occurs in both cases, as shown by the curved shape adopted by the ventral surface (white arrowheads). Apico-basal cables are not visible due to low resolution. Scale bar: 10 μm. An anterior, P posterior, V ventral, D dorsal, Ap apical, B basal. See also Supplementary Figs. 2, 3 and Supplementary Movie 8

Fig. 8g with Fig. 4a, see Supplementary Fig. 4d and Supplementary Movie 9, and compare Fig. 8f, g with 7g, h). Together with the laser dissection experiments, these results strongly suggest that the apico-basal forces generated during apical constriction of EMT committed cells constitute an important driving force in fold formation.

Altogether, our results indicate that EMT constitutes a driving force during morphogenetic processes such as tissue invagination through the generation of a pulling force by delaminating cells.

## Discussion

Apical constriction is generally viewed as one of the main driving forces required to generate epithelium folding[12]. It has been identified in different model systems such as blastopore lip formation in *Xenopus*[13], primitive streak and neural tube folding in chick and mouse[14–17] or mesoderm invagination in *Drosophila*[5,8,18,19] and gives rise to wedge-shaped cells, viewed as a prerequisite for tissue folding. The contribution of apical constriction to folding has been clearly established; nonetheless, an important open question in the field of morphogenesis is the need for additional forces.

Interestingly, apico-basal components have been identified as driving forces in different models of invagination: e.g., apico-basal cell shortening in endoderm invagination in the ascidian[20], or apico-basal force generation by apoptotic cells in *Drosophila* leg folding[11], although the relative importance of apical constriction versus lateral or apico-basal tension was not addressed. In addition to these biological data, a vast majority of the biophysical models developed so far include an apico-basal component (i.e., cell lengthening followed by cell shortening, apico-basal flow, lateral tension), supporting the idea that apical constriction cannot drive mesoderm invagination by itself[11,21–24]. Altogether,

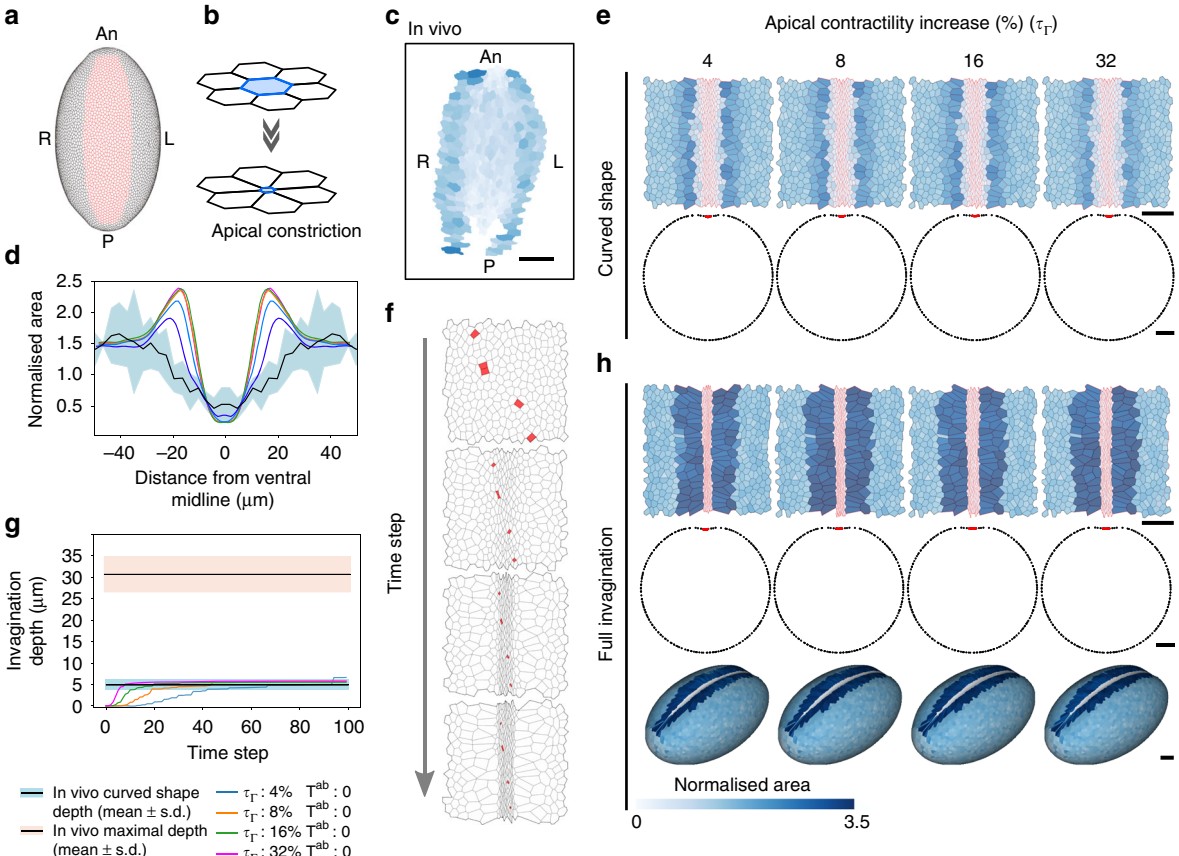

**Fig. 7** Apical constriction is not sufficient to obtain a fold in a 3D biophysical model. **a** Ventral view of the virtual embryo before invagination. Ectodermal cell edges are in black and mesodermal cells in red. An: anterior, P: posterior, L: left, R: right. **b** Scheme of the apical constriction process. The cell constricts its apex (blue) until its surface area falls under a critical threshold. **c** Central region of the curved-shaped mesoderm in vivo. Apical area is color-coded (see color bar below **h**). **d** Comparison of cell area in the curved-shaped mesoderm in vivo and in the model. Graphs show the normalized cell area in vivo (mean: black line, s.d.: blue band, $n = 3$ embryos) and in simulations for different values of apical contractility increase $\tau_\Gamma$ (color-coded as indicated in **g**) with respect to the distance from the ventral midline. **e** Top row: Images show the central region of the curved-shaped mesoderm in simulated embryos for different values of apical contractility increase $\tau_\Gamma$. Apical area is color-coded and cell edges are red in the mesoderm and black elsewhere. Bottom row: Tissue deformation (sagittal views) of the simulation results of curved-shaped embryos. For each simulation, the curved shape was defined as the time step when the invagination depth corresponds to the in vivo depth when the embryo adopts a curved shape. Scale bar: 25 μm. **f** Time-lapse images of the ventral part during mesoderm invagination showing apical extrusion of a few cells (marked in red). **g** Invagination depth as a function of time for different values of apical contractility increase $\tau_\Gamma$ (color-coded as indicated below). The black lines with a blue (resp. pink) border denote the average depth measured in vivo when the embryo adopts a curved (resp. at the end of invagination). **h** Same as (**e**) at full invagination stage for upper and middle rows. Bottom row: Tissue deformation (3D view) of the simulation results at full invagination stage. Scale bar: 25 μm. No apico-basal force was exerted in the above simulations ($T^{ab} = 0$)

these data strongly suggest that other mechanical forces are involved in folding. However, none of these models test directly the relative importance of these forces compared to apical constriction.

Here, using a 3D vertex model, we could directly test the role of apical constriction and found that apical constriction alone only leads to a "curved-shaped" mesoderm. Interestingly, we further showed that including apico-basal traction together with apical constriction is sufficient to mimic ventral furrow formation. Since it has been proposed recently that inducing apical constriction ectopically could be sufficient to induce invagination through the spatial and temporal regulation of Rho activation[25], it would be interesting to test if the generation of an apico-basal force could be a direct consequence of apical constriction. This is the case, for example, for the hydro-dynamic flow identified as a way to transmit apically generated forces deep into the tissue[26], although this flow is not sufficient to drive the full invagination and plays a role only in early events of mesoderm invagination.

Together, these results highlight the importance of apico-basal forces in epithelium folding and suggest that apico-basal forces could be required for tissue remodeling in a wide range of morphogenetic contexts.

Morphogenesis relies mainly on cell rearrangements and the associated cellular forces. If cell division, cell intercalation and cell death are known to participate in tissue remodeling, the potential consequences of EMT on the surrounding tissue have never been characterized. Here, we identify EMT as an actor in morphogenesis that participates non-autonomously in the acquisition of new tissue shapes.

Cells undergoing EMT shift progressively from an epithelial to a mesenchymal state[3]. Starting with apical constriction, the first morphological changes required for basal delamination, and ending with the total loss of epithelial characteristics, cell extrusion and migration outside the epithelial sheet. Curiously, although EMT often coincides with tissue remodeling (e.g., in the chick and mouse primitive streak, the *Xenopus* blastopore lip, the fly ventral furrow), the potential influence of EMT on

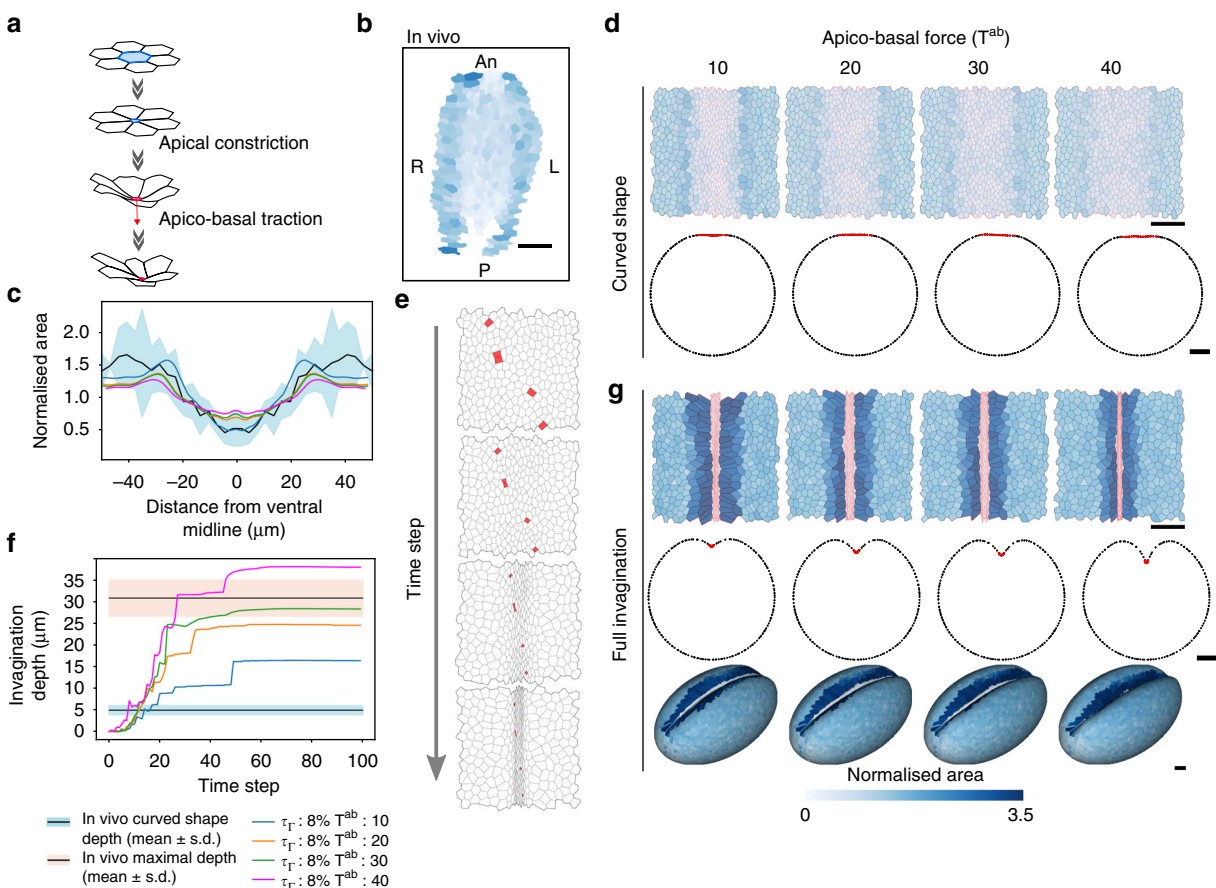

**Fig. 8** Apico-basal traction is required for mesoderm invagination in a 3D biophysical model. **a** Scheme of the apical constriction and the apico-basal traction processes. First, the cell constricts its apex (blue) until its surface area falls under a critical threshold. Then, the cell exerts an apico-basal force (red arrow) perpendicular to the apical surface. **b** Same as Fig. 4c. Central region of the curved-shaped mesoderm in vivo. Apical area is color-coded (see color bar in **g**). **c** Comparison of cell area in the curved-shaped mesoderm in vivo and in the model. Graphs show the normalized cell area in vivo (mean: black line, s.d.: blue band, $n = 3$) and in simulations for different values of apico-basal force $T^{ab}$ (color-coded as indicated in **f**) with respect to the distance from the ventral midline. **d** Top row: Images show the central region of the curved-shaped mesoderm in simulated embryos for different values of apico-basal force $T^{ab}$ and an apical contractility increase rate $\tau_{\Gamma} = 8\%$. Apical area is color-coded and cell edges are red in the mesoderm and black elsewhere (see color bar). Bottom row: Tissue deformation of the modeling results of curved-shaped embryos. For each simulation, curved shape was defined when the invagination depth corresponds to the in vivo depth of curved-shaped embryos. Scale bar: 25 μm. **e** Time-lapse images of the ventral part during mesoderm invagination showing the apical extrusion of a few cells (marked in red). **f** Invagination depth as a function of time for different values of apico-basal force $T^{ab}$ (color-coded as indicated below) and an apical contractility increase rate $\tau_{\Gamma} = 8\%$. The black line with a blue (resp. pink) border denotes the average depth measured in vivo when the embryo adopts a curved shape (resp. at the end of invagination). **g** Same as (**d**) at full invagination stage for upper and middle rows. Bottom: Tissue deformation (3D view) of the simulation results at full invagination stage. Scale bar: 25 μm See also Supplementary Fig. 4 and Supplementary Movie 9

morphogenesis has never been investigated. Indeed, very little is known about the dynamics of the transition and more specifically the influence it could have on the epithelium of origin. To tackle this question, we followed this transition by 3D image analysis in *Drosophila* tissues.

We focused first on the cellular dynamics taking place at the onset of EMT. In the *Drosophila* embryo, we discovered that while constricting their apex, cells from the mesoderm form apico-basal structures of myosin II, here called cables, which generate an apico-basal force. The observation of these specific apico-basal structures, never described so far, may have been possible thanks to the generation of a novel knock-in sqhKI [eGFP] line. Since cables appear preferentially in cells with highly constricted apex, we hypothesize that they form in the pre-delaminating cells as soon as they reach a specific stage in the delamination process. Together with the first phase of sporadic apical constrictions described previously[8,27], this is consistent

with EMT starting first sporadically in a few mesoderm cells, then massively in the whole mesoderm domain, similarly to what has been observed in the chick primitive streak[10]. The same dynamics was found when EMT was induced by ectopic Snail expression, with the formation of an apico-basal cable at the end of apical constriction. In this system we could further visualize a transient deformation of the apical surface of the epithelium at the level of single constricting cells forming an apico-basal myosin II cable, indicating that an apico-basal force was produced.

At the tissue level, the generation of this force by cells undergoing EMT appears essential for morphogenesis. On one hand, ectopic folds form when ectopic EMT is induced; on the other hand, apico-basal forces generated at the onset of EMT appears necessary for mesoderm invagination as shown by the absence of ventral furrow formation when apico-basal forces are prevented. This is reminiscent of lateral constricting forces, which have been predicted more than once in the

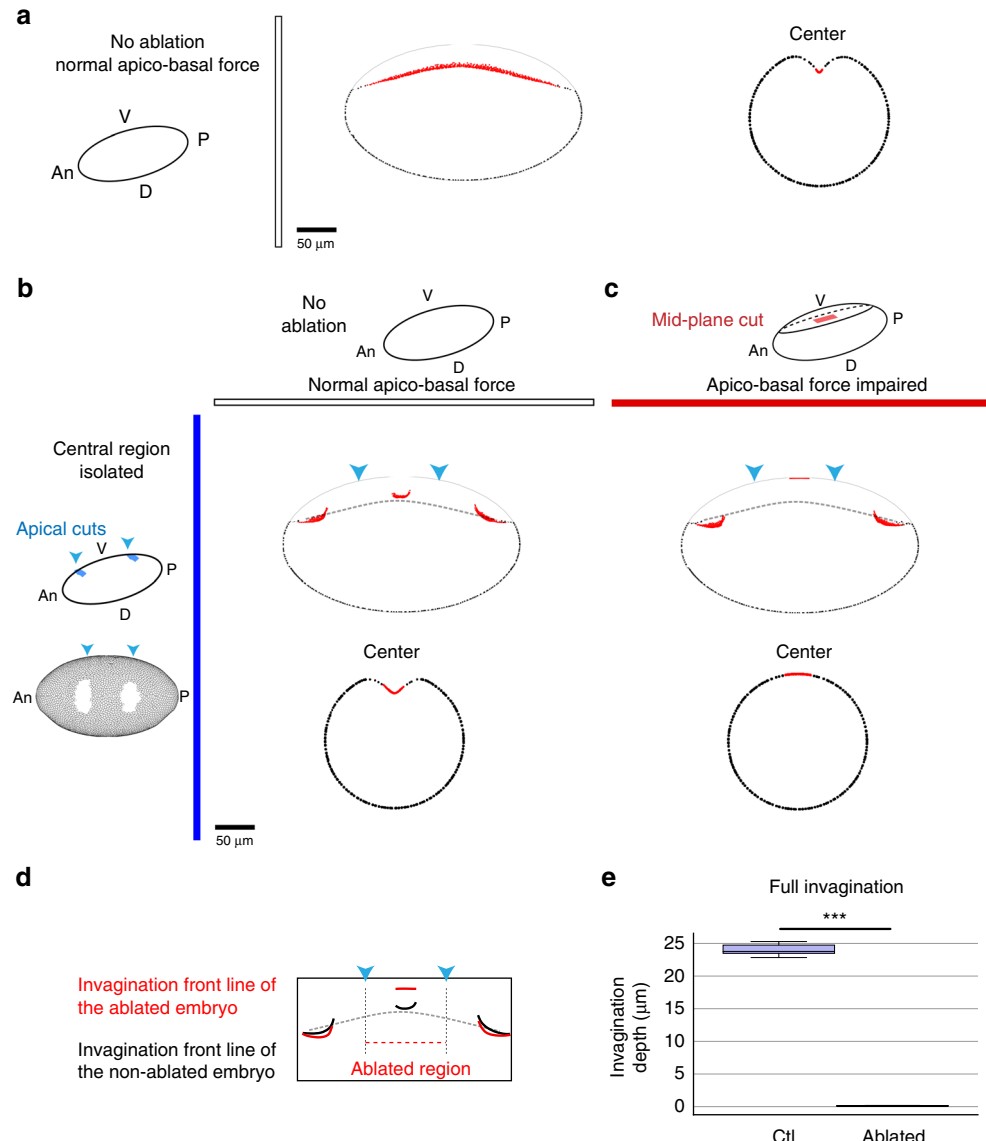

**Fig. 9** Simulating laser ablations in a 3D biophysical model. Schemes recapitulate the different simulated ablations. The apical ablated region is indicated by blue arrowheads. It can be seen on the ventral view of the simulated embryo in b. The mid-plane ablation in (**c**) is represented by a red rectangle in the scheme. Longitudinal and transversal views correspond to time step 50, at this time the invagination is considered as complete; the gray lines represent the initial shape of the embryo. Ectodermal cell edges are in black and mesodermal cells in red. An: anterior, P: posterior, V: ventral, D: dorsal. **a** Longitudinal (left) and transversal (right) sections of the simulated embryo in the control condition. Scale bar: 25 µm. **b**, **c** Longitudinal (top) and transversal (down) sections of the simulated embryos. The dashed gray lines in the transversal sections represent the position of the full invagination profile for the control condition. The invagination is mostly unperturbed in the control condition (**b**, $n = 20$), whereas it is totally prevented when apico-basal forces are impaired (**c**, central region, $n = 20$). Scale bar: 25 µm. **d** Invagination front lines of the simulated embryos shown in (**b**) and (**c**) highlighting the invagination defect in the absence of apico-basal traction compared to the control. **e** Invagination depth of the central region with (ctl) or without (ablated) apico-basal traction, measured respectively when the invagination is maximum. Statistical significance was assessed by the Mann−Whitney test (ctl: $n = 20$ and ablated: $n = 20$ at the end of the invagination: $p = 3.4 \times 10^{-8}$—significant***). See also Supplementary Fig. 5

literature[18,28]. Consistently, ectopic folds form when ectopic EMT is induced.

Altogether, this work reveals that cells entering EMT are not expulsed from the epithelium sheet without consequences on the surrounding epithelial cells, but produce a force orthogonal to the plane of the tissue, both in developmental conditions and in the context of ectopic Snail expression (Fig. 10a). Through this orthogonal force, and thanks to the maintenance of cell−cell adhesion, cells getting ready to delaminate pull on their neighbors and this way participate actively in tissue remodeling and the formation of an invagination (Fig. 10b). These data identify cells

undergoing EMT as key players in tissue morphogenesis and tissue mechanics.

Given that EMT often coincides with tissue folding in developmental contexts[1,2], it will be interesting to test if EMT-driven folding is a general feature of cells undergoing EMT and if this predelamination force is involved in other morphogenetic processes. Furthermore, since EMT is not only an important cellular process recurrently used during development, but is also critical in pathological contexts, EMT being responsible for metastasis formation, the influence of EMT in pathological contexts should be considered in the near future.

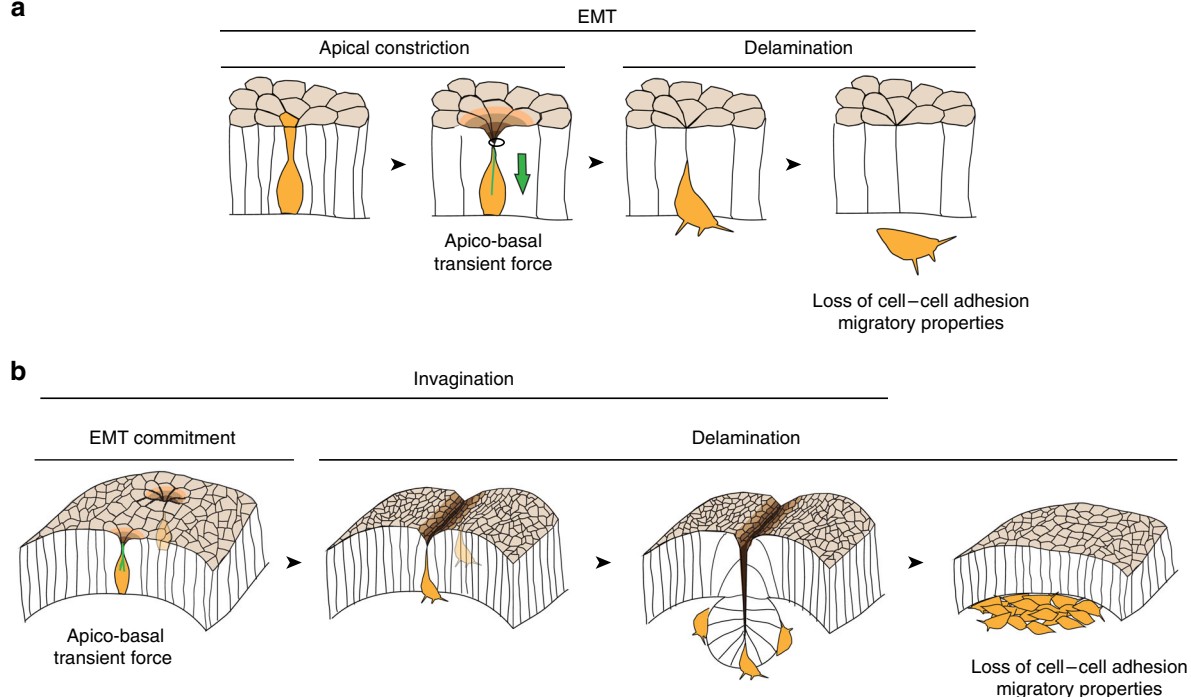

**Fig. 10** Schemes recapitulating our working model on EMT induced folding. **a** Schemes of the successive steps of EMT including apical constriction, generation of an apico-basal pulling force (green arrow) leading to apical deformation, and delamination. **b** Schemes of fold formation driven by the delamination of cells entering EMT. Two cells (yellow) pull transiently on the apical surface of the epithelium, then neighboring cells constrict their apices in this region of the tissue and a fold is formed (depth is color-coded) while cells start to delaminate sporadically, then massively

## Methods

**Fly stocks and genetics**. No ethical approval is required for research projects on *Drosophila*.

In order to respect ethic principles, animals were anesthetized with $CO_2$ (adults) before any manipulation. To avoid any release of flies outside the laboratory, dead flies were frozen before throwing them. Stocks of living flies were conserved in incubators, either at 18, 25 or 30 °C to maintain the flies in optimal condition.

The fluorescent reporters used are the following:

w sqh KI [eGFP] ♯29B and w sqh KI [Tag-RFp-T] ♯3B were designed and generated by InDroso functional genomics (Rennes, France). The respective tags were inserted in C-ter just before the stop codon and the resulting flies were validated by sequencing. The apGal4 line was a gift from Carlos Estella.

uas::α-catenin-TagRFP[29], PH-mCherry (from Y. Bellaiche), UAS::Snail (from J. Kumar) Resille::GFP[30], and sqh[AX3];;sqh::sqhGFP[31] were already described.

Stocks for Snail ectopic expression are: y,w,HS::flp;act>y+>Gal4,uas::GFP (Fig. 1a), sqh::sqh-GFP, UAS::α-catenin-TagRFP/SM5-TM6B/act>CD2>G4 (Fig. 2a–d, f, Supplementary Fig. d–g), sqh KI [eGFP], UAS::α-catenin-TagRFP/SM5-TM6B/ act>CD2>G4 (Figs. 1b, 2g, Supplementary Fig. 1h–i), yw hs::flp; UAS-life-act::Ruby/SM5-TM6B/ act>CD2>G4 (Fig. 1c).

Briefly, the progeny of crosses of interest were grown on standard medium at 25 °C. Third instar larvae were heat shocked for 60 min at 37 °C and dissected between 0 and 2 h APF.

**Immunostainings**. Primary antibodies obtained from Developmental Studies Hybridoma Bank were: rat anti-E-Cad (DCAD2, 1/50) and rat anti-α-Catenin (DCAT-1, 1/50). Rabbit anti-Snail antibody was a gift from Leptin. Secondary antibodies coupled to Alexa-488, -555 and -647 were obtained from Fisher Scientific and diluted 1/200. Samples were mounted in Vectashield with DAPI (Vector Laboratories).

Leg discs from prepupae were dissected in PBS 1×. Tissues are fixed by paraformaldehyde 4% diluted in PBS 1× during 20 min.

Embryos were fixed for 5 min in heptane:formaldehyde 37% (1:1), then devitellinized manually and stained immediately.

After fixation, the samples were washed and saturated in PBS 1×, 0.3% triton x-100 and BSA 1% (BBT). Next, the samples were incubated overnight at 4 °C with primary antibodies diluted in BBT. Samples were washed for 1 h in BBT before a 2 h incubation at room temperature with secondary antibodies diluted in BBT. Finally, samples were washed with PBS 1×, 0.3% Triton x-100 for 1 h and mounted in Vectashields containing DAPI (Vector Laboratories). A 120-μm-deep spacer (Secure-SealTM from Sigma-Aldrich) was placed in between the glass slide and the coverslip to preserve morphology of the tissues.

**Time-lapse imaging**. Leg discs were dissected at white pupal stage in Shields and Sang M3 or Schneider's insect medium (Sigma-Aldrich) supplemented with 15% fetal calf serum and 0.5% penicillin−streptomycin as well as 20-hydroxyecdysone at 2 μg/mL (Sigma-Aldrich, H5142). Leg discs were transferred on a glass slide in 13.5 μL of this medium confined in a 120-μm-deep double-sided adhesive spacer (Secure-SealTM from Sigma-Aldrich). A glass coverslip was then placed on top of the spacer. Halocarbon oil was added on the sides of the spacer to prevent dehydration. Dissection tools were cleaned with ethanol before dissection.

Embryos were dechorionated and mounted in Halocarbon Oil on glue (made by incubating Scotch double-sided adhesive tape overnight in heptane) between a coverslip and a film (Lumox Film 25, ref. 94.6077.317, from Sarstedt).

Imaging was performed using inverted laser scanning confocal microscopes (LSM710 and LSM880 from Zeiss and SP8 from Leica). For rapid imaging of mesoderm invagination, we used inverted spinning disk microscopes (Yokogawa CSU-X1 coupled to Zeiss or Leica microscopes) with either Plan-Apo ×63/1.4 OIL or C-Apo ×63/1.2 Water Autocorr objectives.

**3D reconstruction and deconvolution**. Zen (Zeiss) and Imaris (Bitplane) softwares were used to perform 3D reconstruction and generate sections and 3D projections of tissues. Images were processed with Adobe Photoshop CS5 or ImageJ (https://doi.org/10.1038/nmeth.2019). Deconvolution was performed using the Huygens software to optimize the signal-to-noise ratio.

**Measurements of invagination depth**. Invagination depth was measured with respect to the vitelline membrane in the ventral-most region of the embryo. For the comparison of central and anterior regions, depth was measured in the center of these respective regions.

**Cell segmentation**. Confocal image stacks were preprocessed with Zen (Zeiss) and 3D-median-filtered with ImageJ. Segmentation and shape extraction were performed using MorphoGraphX[32] (http://www.MorphoGraphX.org). First, the images were cut using clipping planes to remove most of the artifact from the acquisition. Then, a solid shape was created to follow the global shape of the embryo by edge detection. A meshwork was generated on this solid shape using the marching cube surface algorithm. The meshwork was smoothed and subdivided three times consecutively and smoothed a last time again. The fluorescence intensity signal was projected on the meshwork before completing the autoseeded morphological watershed algorithm to segment the cells. The segmented polygons were manually corrected for over- or undersegmentation by fusing multiple labels into single cells or dividing one label into multiple cells. Segmentation results were extracted as Polygon File (PLY) Format.

**Constriction rate**. For the quantification of constriction rate of the different row of neighbors from delaminating cells (Fig. 2d), $A_{t+1} - A_t$ ($A$ = cell area) was determined for each time frame (time interval: 7′). The results are presented in box plots in Fig. 2e.

**Quantification of cable-like structures of Myosin II**. Using a 2.5 µm $Z$ projection of the most apical part of the ventral side of an sqhKI[eGFP], Ph-mcherry embryo, we quantified the appearance of new cable-like structure in mesodermal cells for each time point of the first half of the invagination process (the time frame during which apical surface of the cells can be followed accurately).

**Measurement of apical pulses region width**. Apical pulses width was measured in the central part of the ventral-most region of sqhKI[eGFP} embryos for each time frame (time interval: 1′21).

**Statistics**. To assess the differences in invagination depth in different regions of the same embryo (Supplementary Figs. 2c and 5), we paired measurements by embryo and performed the Wilcoxon signed-rank test, considering embryos as independent from each other. The null hypothesis was that differences between paired values were samples from a symmetric distribution centered on 0.

The significance of differences in invagination depth after apical and mid-plane ablation (Supplementary Fig. 2d and Fig. 9) was assessed using the Mann−Whitney test, considering embryos as independent from each other. The null hypothesis was that measurements were samples from the same distribution.

The significance of differences in release after ablation was assessed using the Mann−Whitney test (Fig. 5c), considering embryos as independent from each other. The null hypothesis was that measurements were samples from the same distribution.

The significance of differences in width of myosin II pulses area was assessed using the Mann−Whitney test (Supplementary Fig. 2e). The null hypothesis was that measurements were samples from the same distribution.

Statistics were performed in the R software and significance is denoted as follows according to the $p$ value: ****$p < 0.0001$; ***$p < 0.001$; **$p < 0.01$; *$p < 0.05$; NS: $p \geq 0.05$ (not significant).

Box plots were generated in Excel. The center line represents the median, the upper bound of the box gives the third quartile, the lower bound gives the first quartile and the wiskers give the maximum and minimum values.

**Laser microdissection experiments**. Laser-ablation experiments in embryos were performed with a pulsed DPSS laser ($\lambda = 532$ nm, pulse length = 1.5 ns, repetition rate up to 1 kHz, 3.5 µJ/pulse) steered by a galvanometer-based laser scanning device (DPSS-532 and UGA-42, from Rapp OptoElectronic). The laser beam was focused through an oil-immersion lens of high numerical aperture (Plan-Apochromat ×63/1.4 Imm Oil or LD LCI Plan-Apochromat ×63/1.2 multi-Imm, from Zeiss) at ×0.6 zoom. Photo-disruption was produced in the focal plane.

For ablation experiments, $t0$ was determined using the width of myosin II apical pulses (at $t0$, apical pulses extend between 25 and 38 µm).

For apical cuts, ablation was done following a line of 43 µm (110 px) for 16 s at 70% laser power. For the isolation of the central region of the mesoderm, apical cuts were separated by a distance between 144 and 176 µm. In these conditions, central region was physically separated from the anterior and posterior region by the cuts but otherwise unperturbed. No cauterization occurs in these conditions, so each territory can move freely. The invagination was taking place normally in the central region of control embryos ($n = 10/10$), while it was prevented when mid-plane ablation was performed ($n = 11/12$).

For ablation at 17 µm depth (mid-plane cuts), a rectangle of $33 \times 82$ µm ($85 \times 210$ px) was illuminated for 25 s at 85% laser power. For the measure of apico-basal tension of one specific cable (KI[sqh::GFP]) or lateral membranes (resille::GFP), ablations were performed at 100% laser power following a line of 27 µm (70 px), at around 30 µm depth.

We used an inverted confocal laser scanning microscope (LSM880, from Zeiss) to image live w,sqhKI[eGFP] embryos. The region to be ablated was placed in the center of the field to ensure better reproducibility. 3D image z-stacks were acquired every 81 s.

**Modeling**. For the quasi-static vertex model, we modeled the embryo mesoderm and ectoderm epithelia after cellularization as a mesh of apical junctions, similarly to our previous work[11]. The initial mesh is defined over an ellipsoid with half-axes $2a = 2b = 170$ µm along the left-right and dorsal-ventral axes and $2c = 300$ µm along the anterior-posterior axis, with approximately 6000 cells. Cells are separated in two categories: mesoderm cells and ectoderm cells. The mesoderm is delimited by an elliptic domain with a length of 290 µm and a width of 80 µm, centered on the ventral side of the embryo. It contains 861 cells. All other cells are considered as ectodermal cells. The tissue deforms progressively as mesodermal cells undergo apical constriction through gradual changes in their mechanical parameters, while ectodermal cells passively follow the deformation.

At each iteration, the equilibrium conformation of the apical junction network is given by the minimum of an energy function defined by the following expression:

$$E = \frac{K_Y}{2}(V_Y - V_0)^2 + \sum_\alpha \left( \frac{K_a}{2}(A_\alpha - A_0)^2 + \frac{\Gamma_\alpha}{2}L_\alpha^2 \right) + \sum_i \frac{K_{vit}}{2}\delta\rho_i^2 + \sum_j T_j^{ab} h_j.$$

In this equation, the first term enforces volume conservation of the whole embryo, where $V_Y$ is the volume enclosed by the ellipsoidal apical junction mesh, $V_0$ the preferred volume and $K_Y$ the volume elasticity. The second term is the apical area elasticity of each cell $\alpha$, with $Ka$ the area elasticity, $A\alpha$ the cell area and $A_0$ the preferred cell area. The third term corresponds to cell contractility, where $L\alpha$ denotes the cell perimeter and $\Gamma\alpha$ its contractility. The fourth term constrains the mesh within the vitelline membrane, $K_{vit}$ is the vitelline membrane elasticity and $\delta\rho_i$ is the penetration depth of the vertex $i$ through the vitelline membrane (it is non-null only for vertices contacting the vitelline membrane boundary). The last term models the apico-basal traction and is proportional to the height of the vertex $j$: it is non-null if and only if $j$ belongs to a cell undergoing apical constriction. We consider an anchor point $j'$ as the projection of the vertex $j$ onto the antero-posterior axis. $j'$ is rigidly fixed to this axis. The apico-basal is exerted between $j$ and $j'$. After each iteration, a new energy minimum is searched through a gradient descent strategy using the Broyden−Fletcher−Goldfarb−Shanno bound constrained minimization algorithm from the scipy library[33]. Results are displayed using the Matplotlib (https://doi.org/10.5281/zenodo.1202077) and Ipyvolume libraries (https://doi.org/10.5281/zenodo.1286976).

For cell dynamics in the tissue, we used the following: cells can undergo different processes: apical contractility increase, apico-basal traction, apical relaxation. At each iteration, the respective tasks are stored in an "event manager" in order to be executed at the following iteration. At each iteration, before execution of any task, tasks are shuffled to ensure a random ordering of their execution.

For apical contractility increase, at the first iteration, all mesodermal cells initiate apical constriction: cell contractility $\Gamma_\alpha$ is increased at constant rate $\tau_\alpha$ so that the cell constricts its apex. The contractility increase rate of each cell is defined as $\tau_\alpha = 1 + (\tau_\Gamma - 1)\frac{1+\exp(-kw)}{1+\exp(k(|x_\alpha|-w))}$, where $\tau_\Gamma$ is the maximal contractility value, $k$ is the steepness coefficient characterizing the profile decay, $w$ the width of the profile and $x_\alpha$ the position of the cell $\alpha$ along the left−right axis at the first iteration. During the apical constriction process, when the apical area of the cell $\alpha$ reaches an intermediate threshold, the contractility increase process is propagated to neighboring cells: $\Gamma_\alpha$ is increased by $\tau_\alpha' = \left(\frac{\tau_\alpha - 1.001}{r_{max}}r + \tau_\alpha\right)$, where $r$ is the neighbor order and $r_{max}$ the maximal neighbor order. Cell contractility stops increasing when the apical area falls under a critical threshold $A_c$.

For apico-basal traction modeling, we used the following: The cell can develop an apico-basal tension, during and after the constriction phase, with probability $P = \exp\left(\frac{-A_\alpha}{A_c}\right)$, where $A_\alpha$ is the cell area and $A_c$ is the critical area. The cell is allowed to develop an apico-basal tension only during $N_t$ time steps. Each of the cell's vertices increases its apico-basal tension $T_j^{ab}$ by an equal fraction of $T^{ab}$ so that the cell pulls inwards with a net force of $T^{ab}$.

For apical relaxation, in two ranks of cells at the border between the mesoderm and the ectoderm, cell contractility decreases at a rate $\tau_\Gamma$ and the preferred cell area $A_0$ increases at a rate $\tau_\Gamma$.

To choose parameter values, we applied the following: The unit energy (denoted by $u$) is defined so that the area elasticity modulus $K_a$ equals 1 $u/\mu m^4$. Based on measurements by Lenne and co-workers in the embryo[34], junction stiffness is in the order of $10−50$ pN/µm². In our model, a contractility of 1.12 $u/\mu m^2$ corresponds to a force of approximately 20 $u/\mu m$ for a cell with a typical perimeter of 20 µm. Comparing these two values, $u$ would be in the order of 1 pN/µm. This implies that the apico-basal force amplitude would be about 30 pN, which is consistent with the typical forces generated by acto-myosin fibers[35].

To model yolk incompressibility, yolk volume elasticity $K_Y$ is taken as the lowest value such that apical cell contractility compresses the ellipsoid by less than 1% in volume ($K_\gamma = 3.10^{-6}u/\mu m^6$) (Supplementary Fig. 4a). The initial volume of the yolk is calculated from the dimension of the simulated embryo ($V_0 = 4.56 \times 10^6 \mu m^3$). The vitelline membrane is represented as a rigid external barrier by imposing a high value of $K_{vit}$ and is chosen as 280 $u/\mu m^2$. The width $w$ of the Gaussian curve probability is chosen as 25 µm to obtain an area distribution of cells along the left−right axis that approximates the in vivo experimental results of curved-shaped embryos (Supplementary Fig. 4b). The values of $\tau_\Gamma$ and $T^{ab}$ are chosen so that apical contraction force and apico-basal tension in the mesoderm are of the same order of magnitude ($0.3 < \sum_j T_j^{ab} / \sum_\alpha \Gamma_\alpha L_\alpha < 1.4$). The form of $\tau_\alpha$ as a function of $x_0$ and the steepness $k = 0.19$ are based on myosin activity measurements by Heer et al.[6]. When not specified, the parameter values used for the simulations are those given in this paragraph.

For ablation experiments/simulations, apical and apico-basal cuts were performed only on mesodermal cells. Each set of ablated simulations are based on the same parameters of contractility increase rate and apico-basal traction that best fit the in vivo invagination depth result. It corresponds to a contractility increase rate $\tau_\Gamma = 80\%$ and a net force of $T^{ab} = 30$ $u/\mu m$. All other parameters values are unchanged. Ablations were designed in simulations to mimic ablations performed experimentally.

Apical cuts were modeled by dividing contractility and area elasticity by 100 for the mesodermal cells crossed by a line located 45 μm from the center of the anterior-posterior axis (Fig. 9b, c and Supplementary Fig. 5d, e, blue arrowhead).

For mid-plane cuts, cells in a rectangle of 90 × 80 μm placed at the center of the mesoderm maintain their apico-basal tension to zero (Fig. 9c and Supplementary Fig. 5b, e, red rectangle).

Invagination depth was measured with respect to the vitelline membrane and the corresponding transversal sections (in the center or in the anterior part of the embryo, see figure legends) are presented in Figs. 7–9, Supplementary Figs. 4 and 5.

Morphometric analysis is the same for both the segmented microscopy data and the simulations. The analysis starts with the 3D positions of the cell boundaries. The cell area is determined as the area of the polygon enclosed by the boundary.

Cell areas were measured when the embryo adopts a curved shape, i.e., at the time step when the invagination depth corresponds to the in vivo depth of curved-shaped embryos (4.9 ± 1.1 μm). In order to account for in vivo variability, cell area was normalized to the average cell area at the end of cellularization for each data set.

Mesoderm invagination has been extensively modeled. Most of the models proposed have been in 2D, representing a section of the embryo. The few 3D models developed so far were continuous models. The vertex model developed here has the advantage to take into account the specific dynamic of individual cells (which is a strong limitation in continuous models) and thus is well adapted to mimic the heterogeneity in the timing of apical constriction and delamination that we observed.

**Reporting summary**. Further information on research design is available in the Nature Research Reporting Summary linked to this article.

## Data availability

Results obtained from the vertex model are displayed using the Matplotlib (https://doi.org/10.5281/zenodo.1202077) and Ipyvolume libraries (https://doi.org/10.5281/zenodo.1286976). The data that support all experimental findings of this study are available from the corresponding authors upon reasonable request. Correspondence and requests should be addressed to M.S. and C.B. (magali.suzanne@univ-tlse3.fr and corinne.ben-assayag@univ-tlse3.fr) for biology and materials and to G.G. (guillaume@morphogenie.fr) for modeling. The source data underlying Figs. 2e, 5c, 7d, 8c, f, 9e and Supplementary Figs. 2c−e, 4a−c, 5c, f are provided as a Source Data file.

## Code availability

The code used for modeling is publicly available: https://github.com/suzannelab/invagination.

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

## Acknowledgements

We thank Arnaud Besson, Eric Theveneau, Bruno Monier, Daniela Roellig, Juan Carlos Lopez and Malek Djabali for comments on the manuscript. We thank the Imaging (LITC) and Drosophila facilities of the CBI. M.S.'s lab is supported by grants from the European Research Council (ERC) under the European Union Horizon 2020 research and innovation program (grant number EPAF: 648001), the Fondation Arc pour la Recherche sur le Cancer (CA 09-12-2014) and from the Institut National de la Santé et de la Recherche Médicale (Inserm, Plan cancer 2014–2019). M.G. has been supported by the Ligue Nationale contre le Cancer and the Fondation pour la Recherche Medicale; ST by the ANRT during their Ph.D.

## Author contributions

M.G. conceived and performed the experiments in fly embryos and leg discs, with the help of M.S. for laser-ablation experiments. A.P. set up statistics and participated in modeling conception. G.G. designed the simulation model. S.T. explored the simulation parameters. M.S. supervised the project together with C.B. Funding acquisition: M.S.

## Additional information

**Competing interests:** The authors declare no competing interests.

