## [Peer Review File · Nature Communications]

Reviewers' comments:

Reviewer #1 (Remarks to the Author):

General comments

The manuscript by Gracia and colleagues demonstrates for the first time a role of forces acting orthogonally to the surface of an epithelium during epithelial invagination. The authors explore these forces in two contexts during *Drosophila* development: ectopic expression of the transcription factor Snail in imaginal discs and mesoderm invagination during gastrulation. They show an accumulation of non-muscle Myosin along the lateral side of the cells committed to undergo apical constriction and invagination. To test whether Myosin-driven contractile forces along the apical-basal axis of cells are required for mesoderm invagination, they use laser microdissection targeted to the mid-plane (along the apical-basal axis) of the epithelial cells. This perturbation delays the invagination of the targeted area, and when it is combined with a manipulation that isolates the tissue from the surrounding epithelium tissue invagination is completely blocked. The work is complemented with an *in silico* 3D vertex model that accounts both for superficial (apical) and apical-basal forces during ventral furrow formation, suggesting that apical-basal forces play a part in tissue folding.

The results shown are clear and of high quality and the observations very interesting. They should certainly be published, without any reservations and no further experiments.

But there are a number of issues that need to be addressed, practically all to do with the interpretation, presentation or description of the data, rather than the data themselves. Many are to do with strange or inaccurate use of terms and language.

The most important problem is the term 'EMT', which is used here so loosely or strangely that it would not be wrong to say that title of the manuscript and its main conclusions are overstated. Snail overexpression is not simply equivalent to EMT, the *Drosophila* mesoderm does not undergo EMT during the stages analysed here, and it is even disputable whether the later dispersal of the mesoderm AFTER invagination qualifies as a 'proper' EMT .

The behaviour of individual Snail-expressing cells in the disk epithelium would be much better described as 'delamination', a term that is used here almost interchangeably with 'EMT', which makes nonsense of having two different terms in the first place. Another term that is used is 'extrusion'. This is again problematic because it is a term that has become assigned to processes where cells are 'extruded' from an epithelium because they no longer belong there, for example after cell death or in the context of anoikis. The term 'extrusion' is also used when the apical surface of a cell becomes so small that it can no longer be imaged in a surface view, or perhaps drops below the surface of the rest of the epithelium. But in these cases, the cell may still be within the epithelium, but only at a lower surface. So the cell has NOT necessarily become 'extruded'. The authors absolutely must tidy up their language here, and I would suggest they delete the term EMT from the paper altogether, unless they actually demonstrate that all of the hallmark events of EMT take place in the cases shown here. In the mesoderm, they don't. In the disk epithelium, I suppose we don't know.

The statement that "EMT drives morphogenesis" is also extremely strange. To me, it is in fact meaningless. EMT IS a morphogenetic process. It doesn't drive anything, it just simply IS morphogenesis. It is driven by changes in cell adhesion, polarity, transcriptional changes etc etc., but it doesn't drive anything, except BEING a process of morphogenesis. So the title must be changed. I am not sure why the authors are so obsessed with EMT when they have discovered a really interesting and important process that plays a crucial part in epithelial bending (and has nothing at all to do with EMT!). What I find most surprising is that they don't use their excellent finding to write the paper as stating 'we have finally shown what the Wieschaus lab has predicted more than once (in the old description of cell shortening, and in several more recent papers that postulate a lateral constricting

force, but fail to show it); we now show what they have predicted, and that it has an important role”.

However, important as the force generated by the myosin cables is, and important as the demonstration of their role is, it is simply wrong to make this statement in line 20:
“ cells produce a force, that constitutes an essential driving force for 21 tissue invagination”

It is decidedly not essential, as the experiment in Figure S1 shows: it contributes, but unless apical cuts are made, it is not essential. It is really not necessary to over-hype the language in this manner. ‘essential’ just means something different! Discoveries do not become more important or interesting if the word ‘essential’ is peppered about the manuscript (used ten times - I would guess that every single one could be substituted for ‘important’, or even ‘used’ or similar).

The term cables has been used very loosely by the authors. In the case of actin, cables is used for structures that have periodicity and polarity and one can measure their growth rate as well as stability (localisation of Tropomyosin and ABP). To show if these myosin structures are really “cables”, the authors would have to include FLIP assays and monitor their growth. This is obviously beyond the scope of this work, but maybe it would be wiser to use something like ‘enrichment’ or ‘lateral accumulation’, or at least explain that when they say ‘cables’ this does not imply that there really are cables. They could just be short filaments, for example.

Another use of a term that is unnecessary and irritating is the definition of a stage of gastrulation termed ‘cup stage’. There is nothing resembling a cup here. There is just a ‘furrow’, a term that is well established and understandable. A cup is round, and there is no round indentation. It is also not objectively clear what the ‘cup stage’ is. even seems that the authors refer to ‘cup’ stage for a time point where the furrow is so shallow that it doesn’t even have the curvature of a cup, but is more like a soup plate or even a normal dining plate. What makes it even worse, is that they then use this same term (i.e. a ‘stage’, which is a temporal term) for manipulated embryos in which the timing of invagination is changed (so they are using a morphological term to define a stage in a condition where morphology is changed in timing relative to controls!!)

It would be much better, in fact maybe even necessary, to define precisely what $t = 0$ is for all observations. For example, to choose any morphological event that occurs independent of the manipulations.

A number of specific points, in no particular order:

Figure 3, Laser microdissection experiments.. Chanet et al. (N. Comms 2017) and others (for instance, Martin AC. et al. JCB 2010) suggest that the geometry of the contractile domain influences the outcome of ventral furrow formation. Please indicate the distance between apical cuts in the combined microdissection experiment. Is the distance in the apical cuts are the same between control (apical cuts only) and treatment (apical + mid-section cuts)?

How does the actin-network change in the laser ablation experiments? Would it be possible for the authors to include how these ablations are disrupting the actin localization.

line 96: “At the tissue scale, the force generated by pre-delaminating cells constitutes a driving force for tissue remodeling. Thus ectopic EMT appears sufficient to drive ectopic folding.”
This simply has not been shown by the data at all. This conclusion is not supported (or is meaningless, see above). It has to go.

line 103: “Mesoderm invagination corresponds to the first morphogenetic movement that takes place in the Drosophila embryo.”
Not ‘corresponds to’, it IS the first morphogenetic movement. Why ‘correspond’?

line 128: "some cells start to constrict their apex 128 before the others 6-8."
This was first shown by the Leptin lab in the 1990s, perhaps cite.

Line 111: To identify potential new driving forces generated by cell entering EMT in the mesoderm
No, this is not EMT! They are looking at a stage when there is no sign of EMT at all. For example, all cells maintain their adherens junctions throughout this stage.

Line 150: determining not determinant

Figures in general: The distribution of data over figures in the main paper and supplementary figures is strange. Some of the supplementaries seem totally redundant, while others contain information that is crucial for the main story and should be in the main paper.

Delete 'most' in all 'left-most' 'top-most' etc. Simply wrong in practically all of the cases.

Fig. 1 Delete 'EMT in caption and call it delaminating cells

Fig. 3.

Delete 'essential'.

Add the data from supplementary Fig 2 - they are an integral part of the story and the argument here.

Why are the images on the left a different orientation than the ventral views in the rest of the figure?
(a - p left to right, in the others a - p top to bottom).

line 469: 'measurement of invagination at cup stage' - this doesn't make sense. I thought 'cup stage' was defined as a certain depth of invagination - so how can you measure depth at cup stage? you already know the depth, and if it's a different depth, then it isn't cup stage. this shows how important it is to have an objective and independent time frame.

line116 "wedged-shape" should be wedge-shape

line 765. 'realize' means something else and is wrong here. 'carried out' would work.

Reviewer #2 (Remarks to the Author):

In this manuscript, Gracia and colleagues investigated the mechanisms of tissue folding and proposed that EMT as an important factor to generate the mechanical force in apico-basal direction. The team used 3 systems: *Drosophila* leg imaginal disk, *Drosophila* mesoderm invagination, and in silico vertex model.

First the authors used *Drosophila* leg imaginal disk as a model system and expressed Snail, one of the key factors in EMT. They showed that (i) Snail expressed cells constrict their apical surface and form apico-basal actomyosin cable, and (ii) this cable drives the tissue folding. In mesoderm invagination, the authors found that apico-basal actomyosin cables are formed during mesoderm invagination, especially in the cells with strong apical constriction. This is an interesting observation. By using some sophisticated laser microdissection, the authors demonstrated that the apico-basal actomyosin cables

generate an apico-basal tensile force and are important for mesoderm invagination. The results from in silico 3D vertex model further support the experimental results. Overall, the authors concluded that EMT is a novel factor in tissue morphogenesis (mesoderm invagination).

Although i) I think the question asked and the concept proposed in this manuscript are interesting, and ii) I am in part supportive of this work, there are some crucial questions need to be address. Hope the following suggestions and comments would improve the manuscript.

MAJOR COMMENTS

1) Based on i) *Drosophila* mesoderm is known to undergo EMT at phase 2 of mesoderm invagination, ii) it is known that mesoderm express Snail at phase 1 of mesoderm invagination (this was shown previously), and iii) at phase 1 of mesoderm invagination, some mesoderm cells showed similar characteristics as seen in Snail expressing cells in leg imaginal disk, the authors concluded that apico-basal traction is generated through EMT in phase 1 of mesoderm invagination.

Although these separate observations look logically connected, I feel that some key EMT-characteristics are not clearly shown and are needed to be presented. For instance,

- In mesoderm invagination, the author showed the apico-basal contraction and tissue folding, but not the actual delamination (as shown in Fig. 7b 2nd left). To support the claims by the authors, it is necessary to show that the cells are actually delaminated in phase 1 of mesoderm invagination.

- Other major hallmark of EMT is a reduction of E-cadherin. Although the authors show the adherence junctions remains during the apical constriction, it would be necessarily to show this characteristic of EMT in both Snail cell and mesoderm as an evidence of EMT.

Without some clear evidences of EMT, including delamination and E-cad reduction, I feel that some statements such as "EMT committed cells (line 231)" and "suggests that EMT starts first sporadically in a few mesoderm cells (line 293)" aren't well scientifically supported.

2) How the force transmits from an apico-basal actomyosin cable to the apical part of the cell/tissue? It is clearly shown that an apico-basal actomyosin cable formed and its formation correlated with the tissue folding. Moreover, the 'mid-plane cut' experiment further supported this idea. However, both side of the actomyosin cable need to be anchored to transmit the traction force generated by the apico-basal actomyosin cable to the apical side of the tissue. One side of the cable could be link to the apical section of the cell/tissue. Where the other side of the cable connects to? If the other side of the cable does not connected to anything, the contraction of the actomyosin cable simply ends up with the shortening of the cable, but not tissue folding, i.e., pulling the tissue. As this is a crucial to the overall manuscript, I would suggest the author to look into how the actomyosin cable is anchored and to provide the data.

3) line 163. "This experiment reveals the crucial role of apico-basal forces in mesoderm invagination". Strictly speaking, this statement is correct under the condition with ablation of apical (anterior and posterior) cortex, a.k.a. 'mechanical isolation'. More specifically this statement was supported by the comparison between the experiments shown in Fig 3a and Fig. 3c. Thus the authors should not make this as a general statement and need to specify this special condition (under mechanical isolation) in this statement.

Moreover, if one compares the results between the experiments

[of mid-plane cut without mechanical isolation (Fig. 3c, "invagination was completely impaired (line 161)"), and

[of mid-plane cut with mechanical isolation (Fig. S2b, "laser ablation did not affect tissue development (line 147)"),

a possible conclusion would be 'the apico-basal cable is not essential to drive mesoderm invagination' and 'the apical cortex at apical and posterior section is essential to drive mesoderm invagination'.

I would suggest the authors to take into account this possible alternative interpretation.

MINOR COMMENTS

- 4) I would suggest to highlight in the main text that i) the Snail expression experiments was done in *Drosophila* leg imaginal disk and ii) how you express Snail.
- 5) line 79. "Snail-expressing clones, cells constrict their apex asynchronously". Is this Snail-expressing clones specific? How about the surrounding Snail non-expressing tissue? Are they constrict synchronously? Please quantitatively compare these two cases.
- 6) line 85. "reduce their apical surface more rapidly than the rest of the clone". It is hard to understand this description from the images. It would be helpful to plot the 'rate' of constriction, i.e., $[(A_{t+1}) - (A_t)] / [(t+1) - (t)]$.
- 7) line 137. The statement "essential to drive mesoderm invagination" is misleading as the mesoderm invagination still completes even with mid-plane cut.
- 8) line 138. "Since myosin II cables appear". I feel that it would be helpful to specify this as "apico-basal myosin II cable".
- 9) line 142. "We first set up conditions to specifically target apico-basal myosin cables without affecting the apical pool of myosin II and the subsequent apical constriction". I wish to see a data showing a cut and a recoil of a cable upon laser cut. Having said, I understand this is a difficult experiment. If possible, please provide this data, which would significantly strengthen the manuscript.
- 10) line 150. It is hard to understand the location of the laser cut in Movie S7. It would be helpful to add a line to navigate the readers as the authors did in Fig. S2.
- 11) line 160. I could not see the data nor quantifications of the data showing the "characteristic pulses of apical myosin II". Please provide these data.
- 12) Fig. 1a: It would be helpful to add time stamps to the confocal images.
- 13) I do not think the white dotted rectangle in Fig. 1e represents the image in Fig. 1f'. I would suggest to change the image shown as Fig. 1e.
- 14) Fig. 4g legend. I would suggest to explain what the 4 lines with different colors represent.
- 15) Fig. 5e and legend. It would be helpful to present this in the same way as Fig. 4g.

Reviewer #3 (Remarks to the Author):

The manuscript by Garcia and coworkers revisits what are the forces required to induce tissue invagination during EMT. As an indirect follow up of their previous work in Nature they find that ectopic expression of Snail in the leg imaginal disc induces tissue invagination, combined with the appearance of an apico-basal myosin II cable. They further establish that a similar Myo2 cable can be observed during mesoderm invagination, where Snail normally acts. Laser ablation along the apico-basal axis prevents invagination. This experiment, together with physical vertex modeling of the mesoderm, strongly suggests that this apico-basal force contributes to driving mesoderm invagination during fly embryogenesis and further establishes the importance of EMT as a tissue driving force.

The work will appeal to readers of many fields and is important. Their observation of an apico-basal

Myo2 cable, which had not been missed so far by all previous investigators is the key observation, which was probably been made possible by their generation of a novel Zipper-GFP CRISPR knockin (but it would be important that the authors say a word on this). The experiments are backed by a well-defined vertex model. I strongly recommend publication of this work once several points have been addressed, to a large extent to clarify the results of the laser ablation experiments and the in vivo role of Myo2.

1/ Laser ablation: I don't recall reading about the use of a laser ablation setup relying on a 532 nm ns laser - most so far rely on a UV ns laser (Stelzer version), or infrared ps laser (Lenne version, among others). How is this one comparing? What is the survival rate? Is there any cavitation effect created?

2/ Laser ablation: it was not clear to me how is the embryo mounted for the laser. Is the prospective ventral region facing up such that the laser beam goes through the apico-basal depth of the tissue before reaching 17 μm , and if so how is the region scanned? Or is the laser beam perpendicular to the mid-plane and first going through many cell layers from the lateral side?

3/ Please have a movie to illustrate the effect of the laser opening on either Myo2 or at least on the membrane. The issue here is to ascertain its effect and to demonstrate by other means the occurrence of an apicobasal tension.

4/ The presence of a Myo2 cable together with the physical modeling does suggest that invagination is due to Myo2, but it would be important to further establish this experimentally, especially in light of Stefano de Renzis recent publications showing that apical activation of contractility is sufficient and that basal Myo2 must be down-regulated to observe tissue invagination. I suppose that the best approach would be to use CALI (as previously achieved by the Sanson lab).

Minor

5/ several issues about Fig 4 legend: panel c mentions a color code below panel b which I could not find, the color code for panel d is not mentioned.

6/ Fig S3 legend: instead of "cuts realized", what about "performed".

7/ Movie legends: indicate the time intervals.

8/ Movie S7: I did not understand what is going on in this movie.

9/ Movie S9: could you make a phase diagram.

7/ Fig S5 legend: 2nd line indicates "the mid-plane ablation in c is" but there is no panel c; in panel a' I suppose it is Wilcoxon.

UNIVERSITE PAUL SABATIER

ET

CENTRE NATIONAL DE LA RECHERCHE SCIENTIFIQUE

Laboratoire de Biologie Cellulaire et Moléculaire du Contrôle de la Prolifération
LBCMCP

Magali Suzanne
LBCMCP, CNRS UMR 5088
Université Paul Sabatier, Bât 4R3B1
118 route de Narbonne
31062 TOULOUSE Cedex
email: magali.suzanne@univ-tlse3.fr
tel. 05 61 55 85 88
<http://cbi-toulouse.fr/eng/equipe-suzanne>

Toulouse, 14th of March 2019

Dear referees,

We would like to thank you for your positive and constructive comments. We feel that the results obtained following your suggestions have strengthened the manuscript and we would like to thank you for your time and efforts in considering this work.

Yours sincerely,

Magali Suzanne
Chercheur CNRS (DR2)

Point by point answer to referees comments:

Reviewer #1 :

Major points

1. The most important problem is the term 'EMT, which is used here so loosely or strangely that it would not be wrong to say that title of the manuscript and its main conclusions are overstated. Snail overexpression is not simply equivalent to EMT, the *Drosophila* mesoderm does not undergo EMT during the stages analysed here, and it is even disputable whether the later dispersal of the mesoderm AFTER invagination qualifies as a 'proper' EMT.

The behaviour of individual Snail-expressing cells in the disk epithelium would be much better described as 'delamination', a term that is used here almost interchangeably with 'EMT', which makes nonsense of having two different terms in the first place. Another term that is used is 'extrusion'. This is again problematic because it is a term that has become assigned to processes where cells are 'extruded' from an epithelium because they no longer belong there, for example after cell death or in the context of anoikis. The term 'extrusion' is also used when the apical surface of a cell becomes so small that it can no longer be imaged in a surface view, or perhaps drops below the surface of the rest of the epithelium. But in these cases, the cell may still be within the epithelium, but only at a lower surface. So the cell has NOT necessarily become 'extruded'.

The authors absolutely must tidy up their language here, and I would suggest they delete the term EMT from the paper altogether, unless they actually demonstrate that all of the hallmark events of EMT take place in the cases shown here. In the mesoderm, they don't. In the disk epithelium, I suppose we don't know.

Re: We agree that the EMT was not properly introduced and justified.

We should have demonstrated that all the classical hallmarks of this cellular process were found both in the context of Snail overexpression and in the mesoderm of the embryo.

EMT corresponds to the progressive modifications taking place when an epithelial cell becomes mesenchymal, and is characterized by three hallmarks: the delamination from the epithelial sheet, the progressive loss of cell-cell adhesion, the acquisition of migratory properties.

The full description of these cellular characteristics in a context of Snail overexpression has now been inserted in the new Figure 1 (see below). In our opinion, this demonstrates that the ectopic expression of Snail in the leg discs recapitulate the different characteristics of EMT.

Figure 1. Delamination of Snail expressing cells.

a. Small clones of cells expressing Snail (top: apical view, bottom: sagittal view) in fixed imaginal leg discs, showing the successive steps of cell apical constriction and delamination. Snail-expressing cells are marked by the GFP and outlined in dashed red and epithelium is marked by DE-cadherin in white

Figure 1. Loss of cell adhesion of Snail expressing cells.

b. General view and close-up of big clones of cells expressing Snail (n=19) in fixed imaginal leg discs showing the loss of DE-cadherin in clones that have delaminated and migrate below the epithelium (left frame on the general view and top close-up), while clones inserted in the epithelial sheet keep strong DE-cadherin (right frame in the general view and white arrows in the bottom close-up). DE-cadherin is in white. Apical and basal side of the epithelium are outlined in cyan and yellow respectively.

Figure 1. Migratory properties of Snail expressing cells.

c. Clones of *UAS-life-act::Ruby* (left, control) or *UAS-life-act::Ruby; UAS-Snail* (right) highlighting the migratory properties acquired by Snail expressing cells. Note that the clones (red cells) are well integrated in the monolayer epithelium (outlined in white) in the control (left, *UAS-life-act::Ruby*), while they have delaminated and migrated below the epithelial sheet when Snail is over-expressed (right, *UAS-life-act::Ruby; UAS-Snail*; asterisk) as seen in the general view in b.

In the embryo, EMT is usually described as occurring right after mesoderm invagination, based on the appearance of mesenchymal cells, which corresponds to the final stage of the transition. However, the initiation of the transition has never been described and since the invagination lasts only a few minutes it is tempting to speculate that EMT could start during invagination.

Nevertheless, we agree that visualizing cell delamination would be a strong argument to support this hypothesis. In the previous version of the manuscript, we showed that apical surfaces of a subset of cells totally disappear early in the invagination process and considered this absence of apical surface as the starting point of delamination, a definition that has been used recently (Levayer et al, *Current Biol*, 2016; Marinari et al, *Nature*, 2012). To further support this point, we asked if some cells from the mesoderm were leaving the epithelium before the end of the invagination process. We could show interestingly that although the majority of cells from the mesoderm stay interconnected until the end of the invagination, some cells are extruded during the invagination as shown by the presence of Snail positive cells protruding or extruded from the invaginating epithelial layer (see below). In addition, no DE-cadherin is detected in these extruded cells, suggesting that they have lost their epithelial characteristics. Thus, at least for a subset of mesodermal cells, EMT appears to start earlier than previously anticipated. These data have been incorporated in Figure 3.

c

Fig. 3. Delamination of cells from the mesoderm.

c. Transversal view of PH-mCherry fixed embryos at early (top) or late (bottom) stages of invagination stained with anti-Snail antibody ($n=17$). Note the basal relocation of some nuclei within the mesoderm domain (Snail positive cells, in green) at early stage and the presence of cells at different stage of delamination at later stage: arrowheads point at protruding (white arrowheads) and extruded Snail expressing cells (yellow arrowheads). Note that Snail expression is reduced in extruded cells.

d

Fig. 3. Delamination of cells from the mesoderm.

d. Transversal view of PH-mCherry fixed embryos at late stages of invagination stained with anti-DE-cadherin antibody ($n=11$). Note the presence of protruding (white arrowheads) and extruded cells (yellow arrowheads). DE-cadherin is not detected in extruded cells.

Altogether, considering the different mesenchymal characteristics adopted by cells from the mesoderm and cells over-expressing Snail in the leg disc, we believe that the term EMT is now correctly used.

Finally, Referee 1 suggests that we were using EMT or delamination interchangeably. In fact, the term delamination was used to describe a particular phase of the EMT process corresponding to the extrusion of a cell from the epithelial sheet, and this has been clarified in the new version of the manuscript.

2. The statement that “EMT drives morphogenesis” is also extremely strange. To me, it is in fact meaningless. EMT IS a morphogenetic process. It doesn't drive anything, it just simply IS morphogenesis. It is driven by changes in cell adhesion, polarity, transcriptional changes etc etc., but it doesn't drive anything, except BEING a process of morphogenesis. So the title must be changed. I am not sure why the authors are so obsessed with EMT when they have discovered a really interesting and important process that plays a crucial part in epithelial bending (and has nothing at all to do with

EMT!). What I find most surprising is that they don't use their excellent finding to write the paper as stating 'we have finally shown what the Wieschaus lab has predicted more than once (in the old description of cell shortening, and in several more recent papers that postulate a lateral constricting force, but fail to show it); we now show what they have predicted, and that it has an important role'.

Re: We realize that the statement "EMT drives morphogenesis" was unclear.

Referee 1 probably refers to a cell autonomous aspect of morphogenesis. On the contrary, we focus on the non-autonomous influence of delaminating cells, i.e. the impact they have on the remodeling of the epithelium they are about to leave. This is a novel aspect of delamination that has not been studied as far as we know.

To clarify this point, we changed the title as follow: "Mechanical impact of epithelial-mesenchymal transition on epithelial morphogenesis"

We also modified the text as suggested by the referee (2nd part of the conclusion): "Apico-basal forces generated at the onset of EMT appears necessary for mesoderm invagination as shown by the absence of ventral furrow formation when apico-basal forces are prevented. This is consistent with the existence of lateral constricting force, which has been predicted more than once in the literature (Sweeton et al, 1991; Polyakov et al, 2014)."

3. However, important as the force generated by the myosin cables is, and important as the demonstration of their role is, it is simply wrong to make this statement in line 20:

" cells produce a force, that constitutes an essential driving force for 21 tissue invagination"

It is decidedly not essential, as the experiment in Figure S1 shows: it contributes, but unless apical cuts are made, it is not essential. It is really not necessary to over-hype the language in this manner. 'essential' just means something different! Discoveries do not become more important or interesting if the word 'essential' is peppered about the manuscript (used ten times - I would guess that every single one could be substituted for 'important', or even 'used' or similar).

Re: We agree that "essential" was abusively employed in the previous version of the ms. Most of the time, it has been replaced, as suggested by the referee.

However, we feel that the ablation experiments need some clarifications. Unfortunately, we could not affect apico-basal forces in the whole mesoderm due to technical limitations (due to the curvature of the embryo midplane ablation all along the AP axis of the embryo was not possible without affecting the apical myosin network), however based on the comparison between the three set of experiments, we believe that our conclusion is correct and hope that the referee will agree:

First, the ablation of apico-basal myosin structures in the central part of the embryo leads to a delay of invagination, which indicates that preventing apico-basal forces perturbs the invagination process. Second, the mechanical isolation of the central region by anterior and posterior cuts does not perturb its invagination, which mean that invagination can proceed perfectly normally without the most anterior and posterior regions. Third, the mechanical isolation coupled to the ablation of apico-basal structures of myosin totally prevents invagination, meaning that in this isolated region, which is perfectly able to invaginate normally in the presence of apico-basal cables, apical myosin is not sufficient to drive invagination. Consequently, we showed that in an isolated region that behaves perfectly normally in the presence of apico-basal structures of myosin, midplane ablation is totally deleterious, revealing the importance of apico-basal forces in this invagination process.

We reorganize figure 6 presenting both control invaginations side by side (with of without isolation of the central domain) to show the absence of any perturbation in the invagination speed and depth by apical cuts. This point was not highlighted in the previous version. We hope that this will help clarify this key experiment.

4. The term cables has been used very loosely by the authors. In the case of actin, cables is used for structures that have periodicity and polarity and one can measure their growth rate as well as stability (localisation of Tropomyosin and ABP). To show if these myosin structures are really "cables", the authors would have to include FLIP assays and monitor their growth. This is obviously beyond the scope of this work, but maybe it would be wiser to use something like 'enrichment' or 'lateral accumulation', or at least explain that when they say 'cables' this does not imply that there really are cables. They could just be short filaments, for example.

Re: We agree that the term "cable" has a special meaning and that we do not know here what the organisation of acto-myosin is. We specify in the abstract "the formation of apico-basal structures of myosin II" and use cable later on for simplification. This has been specified now in the text: "this apico-basal accumulation of myosin (hereafter named "cable" for simplification)..."

5. Another use of a term that is unnecessary and irritating is the definition of a stage of gastrulation termed 'cup stage'. There is nothing resembling a cup here. There is just a 'furrow', a term that is well established and understandable. A cup is round, and there is no round indentation. It is also not objectively clear what the 'cup stage' is. even seems that the authors refer to 'cup' stage for a time point where the furrow is so shallow that it doesn't even have the curvature of a coup, but is more like a soup plate or even a normal dining plate. What makes it even worse, is that they then use this same term (i.e. a 'stage', which is a temporal term) for manipulated embryos in which the timing of invagination is changed (so they are using a morphological term to define a stage in a condition where morphology is changed in timing relative to controls!!)

It would be much better, in fact maybe even necessary, to define precisely what $t = 0$ is for all observations. For example, to choose any morphological event that occurs independent of the manipulations.

Re: We agree that "cup stage" was misleading and inadequate, in particular in the case of manipulated embryos, since there is maybe a delay for the acquisition of this specific shape in the mid-plane ablated embryos. It was used initially to describe the initial change in curvature observed at the onset of invagination, before it adopts a V shape. We now call it "curved shape" and explain in the text that it refers to a specific morphology of the embryo, in opposition to "V-shape" invagination, independently of the timing of development.

Regarding the timing of invagination, we define a t_0 in our experiments of laser ablation based on the width of Myosin II apical pulses (see Methods).

Minor points

1. Figure 3, Laser microdissection experiments. Chanet et al. (N. Comms 2017) and others (for instance, Martin AC. et al. JCB 2010) suggest that the geometry of the contractile domain influences the outcome of ventral furrow formation. Please indicate the distance between apical cuts in the combined microdissection experiment. Is the distance in the apical cuts are the same between control (apical cuts only) and treatment (apical + mid-section cuts)?

Re : We agree that the conditions of ablation were not sufficiently detailed.

Consistent with the literature, we noticed that if apical cuts were too close, invagination could be perturbed. However, if the distance between apical cuts was around $160 \mu\text{m}$ ($\pm 16 \mu\text{m}$), invagination was always taking place normally. The distance was the same in both sets of experiments, with or without mid-plane ablations.

We added this information in the Methods section and in the scheme explaining the laser ablation set-up in FigS2:

"For the isolation of the central region of the mesoderm, apical cuts were separated by a distance between 144 and $176 \mu\text{m}$. In these conditions, the invagination was taking place normally in the central

region of control embryos (n=10/10), while it was prevented when mid-plane ablation were performed (n=11/12)”

Figure S2. Laser ablation set-up

a. Midplane ablation set-up: Ablation is performed with a 532nm pulsed laser, through the ventral side of the embryo, in one particular plane located at 17µm deep (cell mid-plane, red line), then a z stack is acquired with the 488nm laser. 10 to 20 cycles of ablation/z imaging are performed, with a time interval of 1'21. Morphogenesis defects are then analyzed on 3D reconstruction images.

b. Isolation of the central domain + midplane ablation set-up: Ablations are performed on the apical surface of the embryo, following a line (bleu line) to isolated the central domain of the mesoderm. Then 10 to 20 cycles of midplane ablations/Z stack imaging are performed, with a time interval of 1'21. Morphogenesis defects are then analyzed on 3D reconstruction images.

2. How does the actin-network change in the laser ablation experiments? Would it be possible for the authors to include how these ablations are disrupting the actin localization?

Re: In order to characterize the impact of laser ablation on the acto-myosin network, we quantified Myosin II apical pulses width and could observe that it is mainly unperturbed over time after apical cuts. These data have been added in FigS2 (see below).

Figure S2. Laser ablation set-up and quantification of the invagination depth

e. Quantification of the width of apical myosin pulses region in control (orange), after apical cuts (blue) and after apical cuts and mid-plane ablations (red) at different time points at the invagination stage. *t*₀ has been defined as described in Methods.

3. line 96: “At the tissue scale, the force generated by pre-delaminating cells constitutes a driving force for tissue remodeling. Thus ectopic EMT appears sufficient to drive ectopic folding.”
This simply has not been shown by the data at all. This conclusion is not supported (or is meaningless, see above). It has to go.

Re: We agree that we did not demonstrate that delaminating cells are responsible for ectopic folding. We modified the text as follow: “At the tissue scale, ectopic EMT appears sufficient to drive ectopic folding suggesting that the force generated by pre-delaminating cells constitutes a driving force for tissue remodeling.”

4. line 103: “Mesoderm invagination corresponds to the first morphogenetic movement that takes place in the *Drosophila* embryo.”
Not ‘corresponds to’, it IS the first morphogenetic movement. Why ‘correspond’?

Re: This has been corrected.

5. line 128: “some cells start to constrict their apex 128 before the others 6-8.”
This was first shown by the Leptin lab in the 1990s, perhaps cite.

Re: The reference has been added.

6. Line 111: To identify potential new driving forces generated by cell entering EMT in the mesoderm
No, this is not EMT! They are looking at a stage when there is no sign of EMT at all. For example, all cells maintain their adherens junctions throughout this stage.

Re: The text has been modified as follow: “To identify new driving forces generated during mesoderm invagination,....” .

7. Line 150: determining not determinant

Re: This has been corrected.

Figures in general: The distribution of data over figures in the main paper and supplementary figures is strange. Some of the supplementaries seem totally redundant, while others contain information that is crucial for the main story and should be in the main paper.

Re: We agree that the organization of the figures was not optimal. They have been reorganized. In particular, the data showing EMT hallmarks of Snail expressing cells are now presented in figure 1 and 3; and the results shown initially in figure S2 (mid-plane ablation of the central region of the embryo) has been insert in figure 6 as suggested below.

8. Delete 'most' in all 'left-most' 'top-most' etc. Simply wrong in practically all of the cases.

Re: This has been corrected.

9. Fig. 1 Delete 'EMT in caption and call it delaminating cells

Re: Figure 1 has been reorganized and is now showing (a) the delamination, (b) the loss of cell adhesion and (c) the migratory properties acquired by Snail expressing cells. It is entitled: Snail over-expression is sufficient to induce ectopic EMT.

10. Fig. 3.

Delete 'essential'.

Add the data from supplementary Fig 2 - they are an integral part of the story and the argument here.

Re: We agree that these results have to be part of the main figures. This has been modified accordingly.

Why are the images on the left a different orientation than the ventral views in the rest of the figure? (a - p left to right, in the others a - p top to bottom).

Re: The figure has been reorganized. The ventral view of the embryo after central domain isolation is now only shown in a supplementary figure S2 in which the ablation set-up is explained (see Referee3 point2).

11. line 469: 'measurement of invagination at cup stage' - this doesn't make sense. I thought 'cup stage' was defined as a certain depth of invagination - so how can you measure depth at cup stage? you already know the depth, and if it's a different depth, then it isn't cup stage. this shows how important it is to have an objective and independent time frame.

Re: We agree that the measurements of cup stage depth was not making sense. They have been removed.

12. line116 "wedged-shape" should be wedge-shape

Re: This has been corrected.

13. line 765. 'realize' means something else and is wrong here. 'carried out' would work.

Re: This has been corrected.

Reviewer #2 :

Major points

1) Based on i) *Drosophila* mesoderm is known to undergo EMT at phase 2 of mesoderm invagination, ii) it is known that mesoderm express Snail at phase 1 of mesoderm invagination (this was shown previously), and iii) at phase 1 of mesoderm invagination, some mesoderm cells showed similar characteristics as seen in Snail expressing cells in leg imaginal disk, the authors concluded that apico-basal traction is generated through EMT in phase 1 of mesoderm invagination.

Although these separate observations look logically connected, I feel that some key EMT-characteristics are not clearly shown and are needed to be presented. For instance,

- In mesoderm invagination, the author showed the apico-basal contraction and tissue folding, but not the actual delamination (as shown in Fig. 7b 2nd left). To support the claims by the authors, it is necessary to show that the cells are actually delaminated in phase 1 of mesoderm invagination.
- Other major hallmark of EMT is a reduction of E-cadherin. Although the authors show the adherence junctions remains during the apical constriction, it would be necessarily to show this characteristic of EMT in both Snail cell and mesoderm as an evidence of EMT.
- Without some clear evidences of EMT, including delamination and E-cad reduction, I feel that some statements such as “EMT committed cells (line 231)” and “suggests that EMT starts first sporadically in a few mesoderm cells (line 293)” aren’t well scientifically supported.

Re: We agree that data were missing to support some of our statements.

Cell delamination

As discussed in the response to Referee 1 point1, EMT is usually described as occurring after mesoderm invagination, based on the time at which cells have lost their epithelial characteristics and acquired mesenchymal ones, which corresponds to the final stage of the transition. However, the initiation of the transition has never been described. Furthermore, the invagination lasts a few minutes and cells are mesenchymal as soon as the invagination ends, which strongly suggests that EMT starts during invagination.

Nevertheless, we agree that visualizing cell delamination would be a strong argument to support this hypothesis. In the previous version of the manuscript, we showed that apical surfaces of a subset of cells totally disappear early in the invagination process and considered this absence of apical surface as the starting point of delamination, a definition that has been used recently (Levayer et al, Current Biol, 2016; Marinari et al, Nature, 2012). To further support this point, we looked if some cells from the mesoderm were leaving the epithelium before the end of the invagination process. We could show that although the majority of cells from the mesoderm stay interconnected until the end of the invagination, some cells are extruded during the invagination as shown by the presence of Snail expressing cells protruding or extruded from the invaginating epithelial layer (see below, arrowheads). Although these events are rare, which explain why they have not been described before, this indicates that some delaminations do take place during invagination. Thus, at least for a subset of mesodermal cells, EMT appears to start earlier than previously anticipated. These data have been incorporated in Figure 4 (see below).

c

Fig. 3. Delamination of cells from the mesoderm.

c. Transversal view of PH-mCherry fixed embryos at early (top) or late (bottom) stages of invagination stained with anti-Snail antibody ($n=17$). Note the basal relocation of some nuclei within the mesoderm domain (Snail positive cells, in green) at early stage and the presence of cells at different stage of delamination at later stage: arrowheads point at protruding (white arrowheads) and extruded Snail expressing cells (yellow arrowheads). Note that Snail expression is reduced in extruded cells.

Loss of cell-cell adhesion:

In addition, the description of cell-cell adhesion loss, another hallmark of EMT, was missing in the previous version of the manuscript.

In the mesoderm, the loss of E-cadherin has been described when cells have delaminated and form an inner layer of mesenchymal cells (Schäfer et al, JCS, 2014). To figure out how E-cadherin was distributed in extruding cells, we looked at E-cadh distribution during invagination. We could observed that no E-cadherin is detected in protruding or extruded cells, strongly suggesting that they have lost their epithelial characteristics. These new data have been inserted in the new Figure 4 (see below).

d

Fig. 3. Delamination of cells from the mesoderm.

d. Transversal view of PH-mCherry fixed embryos at late stages of invagination stained with anti-DE-cadherin antibody ($n=11$). Note the presence of protruding (white arrowheads) and extruded cells (yellow arrowheads). DE-cadherin is not detected in extruded cells.

We did the same experiments in Snail over-expressing cells and could show that E-cadherin is lost in delaminating cells. These data are now included in Fig1 in the new version of the manuscript (see below).

Figure 1. Loss of cell adhesion of Snail expressing cells.

b. General view and close-up of big clones of cells expressing Snail (n=19) in fixed imaginal leg discs showing the loss of DE-cadherin in clones that have delaminated and migrate below the epithelium (left frame on the general view and top close-up), while clones inserted in the epithelial sheet keep strong DE-cadherin (right frame in the general view and white arrows in the bottom close-up). DE-cadherin is in white. Apical and basal side of the epithelium are outlined in cyan and yellow respectively.

Altogether, considering the different mesenchymal characteristics adopted by cells from the mesoderm and cells over-expressing Snail in the leg disc, we believe that the term EMT is now correctly used.

[REDACTED]

3) line 163. “This experiment reveals the crucial role of apico-basal forces in mesoderm invagination”. Strictly speaking, this statement is correct under the condition with ablation of apical (anterior and posterior) cortex, a.k.a. ‘mechanical isolation’. More specifically this statement was supported by the comparison between the experiments shown in Fig 3a and Fig. 3c. Thus the authors should not make this as a general statement and need to specify this special condition (under mechanical isolation) in this statement.

Moreover, if one compares the results between the experiments [of mid-plane cut without mechanical isolation (Fig. 3c, “invagination was completely impaired (line 161)”)], and [of mid-plane cut with mechanical isolation (Fig. S2b, “laser ablation did not affect tissue development (line 147)”)], a possible conclusion would be ‘the apico-basal cable is not essential to drive mesoderm invagination’ and ‘the apical cortex at apical and posterior section is essential to drive mesoderm invagination’. I would suggest the authors to take into account this possible alternative interpretation.

Re: As mentioned to Referee 1, we feel that the ablation experiments needed some clarifications. Unfortunately, we could not affect apico-basal forces in the whole mesoderm due to technical limitations (due to the curvature of the embryo midplane ablation all along the AP axis of the embryo was not possible without affecting the apical myosin network), however based on the comparison between the three set of experiments, we believe that our conclusion is correct and hope that the referee will agree.

First, the ablation of apico-basal myosin structures in the central part of the embryo leads to a delay of invagination, which indicates that preventing apico-basal forces perturbs the invagination process.

Second, the mechanical isolation of the central region by anterior and posterior cuts does not perturb its invagination, which means that invagination can proceed perfectly normally without the most anterior and posterior regions. Third, the mechanical isolation coupled to the ablation of apico-basal structures of myosin totally prevents invagination, meaning that in this isolated region, which is perfectly able to invaginate normally in the presence of apico-basal cables, apical myosin is not sufficient to drive invagination. Consequently, we showed that in an isolated region that behaves perfectly normally in the presence of apico-basal structures of myosin, midplane ablation is totally deleterious, revealing the importance of apico-basal forces in this invagination process.

Comparing mid-plane ablation with and without isolation of the central domain does not allow to conclude on the importance of apico-basal forces, however, it shows that forces generated in the anterior and posterior domains are sufficient to drag the central region inside.

These conclusions have been clarified in the text:

“Altogether, these experiments reveal (1) that the ablation of apico-basal structures in the central part of the embryo leads to a delay of invagination, which indicates that preventing apico-basal forces perturbs the invagination process; (2) that the mechanical isolation of the central region by anterior and posterior cuts does not perturb its invagination, which means that invagination can proceed normally without the most anterior and posterior regions; (3) that the mechanical isolation coupled to the ablation of apico-basal structures totally prevents invagination, indicating that in the isolated region, which is perfectly able to invaginate normally in the presence of apico-basal forces, apical myosin is not sufficient to drive invagination and (4) midplane ablation is totally deleterious. Altogether, these experiments reveal the crucial role of apico-basal forces in mesoderm invagination.”

Minor points

4) I would suggest to highlight in the main text that i) the Snail expression experiments was done in *Drosophila* leg imaginal disk and ii) how you express Snail.

Re: This has been corrected.

5) line 79. “Snail-expressing clones, cells constrict their apex asynchronously”. Is this Snail-expressing clones specific? How about the surrounding Snail non-expressing tissue? Are they constrict synchronously? Please quantitatively compare these two cases.

Re : Indeed, apical constriction is specific to Snail expressing cells. Quantifications of the apical constriction rate in Snail expressing clones clarifies this point (see point 6 below).

Apical surface of Snail expressing (red) and non-expressing cells, visualized by DE-cadherin staining. Note that none of Snail non-expressing cells constrict their apex, compared to the Snail expressing cells, which constrict their apex asynchronously as shown by the heterogeneity of apical surface in the red domain.

6) line 85. “reduce their apical surface more rapidly than the rest of the clone”. It is hard to understand this description from the images. It would be helpful to plot the ‘rate’ of constriction, i.e., $[(A_{t+1}) - (A_t)] / [(t+1) - (t)]$.

Re : We agree that this point was not clear. We modified the figure, color coding the 1st, 2nd, 3rd and 4th row of neighbors of the constricting cell and quantify their respective rate of constriction, which highlight the gradual effect of the pulling cells on its neighbors. This has been included in Fig2 (see below).

Figure 2. Cells undergoing EMT generate an apico-basal force actively involved in tissue folding
d. Time-lapse images of a Snail expressing clone. 1st(yellow) , 2nd (green) , 3rd (orange) and 4th (bleu) row of neighbors from the delaminating cell (red) are color coded to highlight that cells in close vicinity to the delaminating cell have a higher apical constriction rate (apical views). The whole tissue gradually changes shape (outlined in red, sagittal view), ultimately producing a stable invagination (white arrow). See also Movie S2.
e. Quantification of apical constriction rate in 1st(yellow) , 2nd (green) , 3rd (orange) and 4th (bleu) row of neighbors from the delaminating cell (see Methods).

7) line 137. The statement “essential to drive mesoderm invagination” is misleading as the mesoderm invagination still completes even with mid-plane cut.

Re: We hope our response to point 3 clarify this point.

8) line 138. “Since myosin II cables appear”. I feel that it would be helpful to specify this as “apico-basal myosin II cable”.

Re: This has been corrected.

9) line 142. “We first set up conditions to specifically target apico-basal myosin cables without affecting the apical pool of myosin II and the subsequent apical constriction”. I wish to see a data showing a cut and a recoil of a cable upon laser cut. Having said, I understand this is a difficult experiment. If possible, please provide this data, which would significantly strengthen the manuscript.

Re: We agree with the referee that the demonstration of an apico-basal tension was missing in the previous version of the manuscript.

As mentioned to Referee1 (minor point 2), we analyze the effect of ablation on the acto-myosin cytoskeleton using a tagged myosin light regulatory chain of sqh (knock-in construct generated in the lab). Although challenging, we eventually set up the conditions to optimize the chance to visualize a myosin II cable in one particular focal plane by orienting the embryo laterally, with the ventral part slightly tilted toward the coverslip. By this way, we succeed to cut specifically these transient myosin structures and compare the tension borne by these cables to the lateral tension present in the rest of the epithelium. As shown below, the cable is clearly under tension and its cut results in a recoil along the apico-basal axis (n=11/11), compared to control ablation of the membrane in the mid-plane of the epithelium (n= 11/11). These results are included in the new figure 5 (see below).

Figure 5. Apico-basal cables formed in mesoderm cells are under tension.

a. Laser ablation of apico-basal myosin cable. (top) Images extracted from a movie and corresponding schemes showing transversal views of a KI-sqh::GFP embryo mesoderm cells showing a myosin cable before (red bracket), during (black) and after (green bracket) laser cut (red dotted line). In the kymograph (bottom), the timing of laser cut is indicated by the red line, prolonged by a dotted red line to visualize the site of ablation, and the cable release highlighted by yellow dotted line. Blue arrows indicate the basal release.

b. Control laser ablation of lateral membranes of the embryo (resille::GFP line). Images extracted from a movie and corresponding schemes showing transversal views of the epithelium before, during and after laser ablation. The laser cut is indicated with a red dotted line. Note that the fluorescence at the membranes appear to recover rapidly.

c. Curves of the average recoil observed between the region of ablation and the remaining cable (as shown by the blue arrow in the kymograph) or the remaining membrane. Statistical significance was assessed by the Wilcoxon signed-rank test (control cut (membrane): $n=11$ embryos; apico-basal cable cut: $n=11$).

10) line 150. It is hard to understand the location of the laser cut in Movie S7. It would be helpful to add a line to navigate the readers as the authors did in Fig. S2.

Re: We agree with the referee that the defect is too subtle to be visible in a movie. We choose to present still images of the movie instead, focusing on the stage at which the invagination delay can be seen in the central domain. These data are now presented in figure 6:

Figure 6. An apico-basal force drives mesoderm invagination.

a. Longitudinal views of *sqhKI[eGFP]* control embryos ($n=17$), showing the linear shape of the invagination front at early stage.

b. (top) Longitudinal views of *sqhKI[eGFP]* embryo ablated in the midplane region (red rectangle in the scheme and a dashed red line in the figure, $n=12$), showing a slight delay in the central region compared to the anterior one, as shown by the outline of the invagination front in red at early stage of invagination. (bottom) Scheme of the invagination fronts in the control (white line, as in a) and ablated embryo (red line).

b'. Anterior (left) and central (right) transversal views of a midplane ablated embryo, at two different time of the invagination. Note that invagination proceeds normally as shown by the V-shaped invagination outlined by a white dotted line (bottom), although a delay in the central region can be observed at early stage (top, compare anterior to central region).

11) line 160. I could not see the data nor quantifications of the data showing the “characteristic pulses of apical myosin II”. Please provide these data.

Re : We agree that quantification should be added to convince that apical myosin was unperturbed. In order to characterize the impact of laser ablation on the apical acto-myosin network, we quantified Myosin II apical pulses width and could observe that it is mainly unperturbed over time after apical cuts. These data have been added in FigS2 (see below).

Figure S2. Laser ablation set-up and quantification of the invagination depth
e. Quantification of the width of apical myosin pulses region in control (orange), after apical cuts (blue) and after apical cuts and mid-plane ablations (red) at different time points at the invagination stage. *t0* has been defined as described in Methods.

12) Fig. 1a: It would be helpful to add time stamps to the confocal images.

Re: The images presented in figure 1a are not extracted from a movie, but are a reconstruction of the different step of delamination from fixed samples. We are sorry for the confusion and modify the legend accordingly.

13) I do not think the white dotted rectangle in Fig. 1e represents the image in Fig. 1f'. I would suggest to change the image shown as Fig. 1e.

Re: The image has been changed.

14) Fig. 4g legend. I would suggest to explain what the 4 lines with different colors represent.

Re: This has been corrected.

15) Fig. 5e and legend. It would be helpful to present this in the same way as Fig. 4g.

Re: This has been corrected.

Reviewer #3:

Their observation of an apico-basal Myo2 cable, which had not been missed so far by all previous investigators is the key observation, which was probably been made possible by their generation of a novel Zipper-GFP CRISPR knockin (but it would be important that the authors say a word on this).

Re: Indeed, this is an important point that we now mentioned in the manuscript:

I.322: "The observation of these specific apico-basal structures, never described so far, may have been possible thanks to the generation of a novel knock-in sqhKI[eGFP] line."

Major points

1/ Laser ablation: I don't recall reading about the use of a laser ablation setup relying on a 532 nm ns laser - most so far rely on a UV ns laser (Stelzer version), or infrared ps laser (Lenne version, among others). How is this one comparing? What is the survival rate? Is there any cavitation effect created?

Re: Although it is true that 532nm ns laser is not the most commonly used in the literature, it has already been used for ablation without showing any more side effect than UV ns or infrared ps ones (Monier et al., Nature,2015; Taty-Taty et al., Cell Cycle, 2014; Courtheoux et al, JCB, 2009; Cabello et al, JCB, 2016). The use of this laser does not affect survival rate of the embryos as judged by their normal development after mid-plane ablation alone, both in term of morphology and timing. Furthermore, myosin dynamics is normally accumulated at the membrane of ectodermal cells and neuroblasts divide normally after invagination is complete.

Finally, no cavitation is observed as shown when ablation is performed on the membrane of Resille-GFP embryos in the exact same conditions used for Myosin II cables ablations (see point 3 below). Importantly, a 532nm pulsed laser has the advantage to go deeper in the tissue compared to UV, which was an important parameter for us since ablation was performed at 17µm to 30µm depth (figure 5 and 6), and can be used with objectives used for visible wave lengths, which is not the case for infrared and could be combined with optimal fluorescent imaging. Finally, the repetition rate is low (up to 1kHz) avoiding heating and cavitation.

2/ Laser ablation: it was not clear to me how is the embryo mounted for the laser. Is the prospective ventral region facing up such that the laser beam goes through the apico-basal depth of the tissue before reaching 17 µm, and if so how is the region scanned? Or is the laser beam perpendicular to the mid-plane and first going through many cell layers from the lateral side?

Re: We agree that the setup was not clearly explained. The laser beam is going through the apico-basal depth of the tissue, the incidence of the laser facing the ventral region of the embryo. It is important to note that the laser is a bi-photon, and ablation is restricted to the focal plane. We now provide a scheme of laser ablation setup in FigS2 to clarify this point (see below).

Figure S2. Laser ablation set-up

a. Midplane ablation set-up: Ablation is performed with a 532nm pulsed laser, through the ventral side of the embryo, in one particular plane located at 17µm deep (cell mid-plane, red line), then a z stack is acquired with the 488nm laser. 10 to 20 cycles of ablation/z imaging are performed, with a time interval of 1'21. Morphogenesis defects are then analyzed on 3D reconstruction images.

b. Isolation of the central domain + midplane ablation set-up: Ablations are performed on the apical surface of the embryo, following a line (bleu line) to isolated the central domain of the mesoderm. Then 10 to 20 cycles of midplane ablations/Z stack imaging are performed, with a time interval of 1'21. Morphogenesis defects are then analyzed on 3D reconstruction images.

3/ Please have a movie to illustrate the effect of the laser opening on either Myo2 or at least on the membrane. The issue here is to ascertain its effect and to demonstrate by other means the occurrence of an apicobasal tension.

Re : We agree with the referee that the demonstration of an apico-basal tension was missing in the previous version of the manuscript.

As mentioned to Referee2 (minor point 9), we analyze the effect of ablation on the acto-myosin cytoskeleton using a tagged myosin light regulatory chain of sqh (knock-in construct generated in the lab). Although challenging, we eventually set up the conditions to optimize the chance to visualize a myosin II cable in one particular focal plane by orienting the embryo laterally, with the ventral part slightly tilted toward the coverslip. By this way, we succeed to cut specifically these transient myosin structures and compare the tension borne by these cables to the lateral tension present in the rest of the epithelium. As shown below, the cable is clearly under tension and its cut results in a recoil along the apico-basal axis (n=11/11), compared to control ablation of the membrane in the mid-plane of the epithelium (n= 11/11). These results are included in the new figure 5 (see below).

Figure 5. Apico-basal cables formed in mesoderm cells are under tension.

a. Laser ablation of apico-basal myosin cable. (top) Images extracted from a movie and corresponding schemes showing transversal views of a *KI-sqh::GFP* embryo mesoderm cells showing a myosin cable before (red bracket), during (black) and after (green bracket) laser cut (red dotted line). In the kymograph (bottom), the timing of laser cut is indicated by the red line, prolonged by a dotted red line to visualize the site of ablation, and the cable release highlighted by yellow dotted line. Blue arrows indicate the basal release.

b. Control laser ablation of lateral membranes of the embryo (*resille::GFP* line). Images extracted from a movie and corresponding schemes showing transversal views of the epithelium before, during and after laser ablation. The laser cut is indicated with a red dotted line. Note that the fluorescence at the membranes appear to recover rapidly.

c. Curves of the average recoil observed between the region of ablation and the remaining cable (as shown by the blue arrow in the kymograph) or the remaining membrane. Statistical significance was assessed by the Wilcoxon signed-rank test (control cut (membrane): $n=11$ embryos; apico-basal cable cut: $n=11$).

4/ The presence of a Myo2 cable together with the physical modeling does suggest that invagination is due to Myo2, but it would be important to further establish this experimentally, especially in light of Stefano de Renzis recent publications showing that apical activation of contractility is sufficient and that basal Myo2 must be down-regulated to observe tissue invagination. I suppose that the best approach would be to use CALI (as previously achieved by the Sanson lab).

Re: We agree that the use of the CALI technique would have been a nice complementary experiment to further test the role of Myosin II in apico-basal force generation. However, this approach, although efficient to locally inactivate Myosin II at the apical surface of the embryo as described by the lab of Benedicte Sanson (Monier et al, 2010), proved inefficient in the depth of the tissue (even at full laser power). This totally excludes the possibility of using this technique for apico-basal Myosin II structures.

Minor points

5/ several issues about Fig 4 legend: panel c mentions a color code below panel b which I could not find, the color code for panel d is not mentioned.

Re: This has been corrected.

6/ Fig S3 legend: instead of "cuts realized", what about "performed".

Re: This has been corrected.

7/ Movie legends: indicate the time intervals.

Re: This has been added.

8/ Movie S7: I did not understand what is going on in this movie.

Re: We agree with the referee that the defect is too subtle to be visible in a movie. We choose to present still images of the movie instead, focusing on the stage at which the invagination delay can be seen in the central domain. These data are now presented in figure 6 (see below).

Figure 6. An apico-basal force drives mesoderm invagination.

a. Longitudinal views of *sqhKI[eGFP]* control embryos ($n=17$), showing the linear shape of the invagination front at early stage.

b. (top) Longitudinal views of *sqhKI[eGFP]* embryo ablated in the midplane region (red rectangle in the scheme and a dashed red line in the figure, $n=12$), showing a slight delay in the central region compared to the anterior one, as shown by the outline of the invagination front in red at early stage of invagination. (bottom) Scheme of the invagination fronts in the control (white line, as in *a*) and ablated embryo (red line).

b'. Anterior (left) and central (right) transversal views of a midplane ablated embryo, at two different time of the invagination. Note that invagination proceeds normally as shown by the V-shaped invagination outlined by a white dotted line (bottom), although a delay in the central region can be observed at early stage (top, compare anterior to central region).

9/ Movie S9: could you make a phase diagram.

Re: Since there is no clear demarcation between a domain where a fold forms and a domain where the tissue stays flat, but rather a continuous increase in the fold depth as a function of apico-basal force and apical contractility, a phase diagram can not be drawn.

7/ Fig S5 legend: 2nd line indicates "the mid-plane ablation in c is" but there is no panel c; in panel a' I suppose it is Wilcoxon.

Re: This has been corrected. We apologize for the mistake.

REVIEWERS' COMMENTS:

Reviewer #1 (Remarks to the Author):

This was a very good paper already, and my comments were minor. The authors have dealt with them in a satisfactory manner. This work should go out as soon as possible.

Reviewer #2 (Remarks to the Author):

The authors adequately addressed all of my comments. The data newly added strengthened this manuscript significantly. Now I support this revision for publication after addressing the following minor comments.

Fig. 2e: Statistical tests are missing.

Fig. 3d: It would be helpful to outline the shape of the cells of interests (highlighted by arrows) by dotted line.

Reviewer #3 (Remarks to the Author):

The revised version of the manuscript by Gracia and colleagues is much improved and has addressed my comments, and as far as I could tell those made by the other two reviewers. In particular, the new experiments reported in Fig 5 establish that the actin cables are under tension and those in Fig 6 support the notion that those cables are essential for mesoderm invagination. I was also convinced by the new data reported in Figs 1&2. So altogether I reiterate my very strong support for publication of this work in Nature Communications.

I have a few minor comments:

1. Line 84: we could observe (no d)
2. Line 135: disappear (two p)
3. Line 147: add "novel" before knock-in. By the way, I apologize for not properly proofreading my original review in which I had written "Their observation of an apico-basal Myo2 cable, which had not been missed so far by all previous investigators is the key observation," when I meant "which had been missed" and I hope it was clear to everyone.
4. I find that the experiment in which they "isolate the central region" not properly described even in the Methods section. I wasn't entirely clear to me whether the authors prevented apical pulses in the anterior and posterior regions, fried the prospective mesoderm cells in those regions, or glued them to the yolk as previously done by Collinet and Lecuit. Please clarify.
5. Line 286: the word "alternative" is both badly chosen to describe what they mean and perhaps tainted these days (alternative facts....). I suggest that you write "the need for additional forces".
6. Figure 1 & 2 legends panel a: the legend says small clones of cells expressing Snail in "fixed imaginal discs" or "generated in fixed imaginal discs". Taken at face value, it looks contradictory to generate a clone in a fixed sample. So, I imagine that the authors mean that they are showing images taken from fixed samples in which they had generated clones prior to fixation. Please clarify.
7. A somewhat general comment about actin cables. Taking together Figs. 4a, 4a', 4c, 6c, it wasn't clear to me what is the distribution of actin cables in the prospective mesoderm as it undergoes invagination/EMT. Could you please quantify as to what is the distribution (1/cell, 1 every few cell). If the distribution is significantly below 1/cell, is the model described in Fig 8 still compatible and considering that possibility?
8. In the modeling section of the Methods, add a few lines to contrast this new model to the many

models that have been made to account for mesoderm invagination.
9. Please provide a video corresponding to 5a.

Point by point answer to referees comments:

Reviewer #1 (Remarks to the Author):

This was a very good paper already, and my comments were minor. The authors have dealt with them in a satisfactory manner. This work should go out as soon as possible.

Re: We would like to thank you for your positive comments.

Reviewer #2 (Remarks to the Author):

The authors adequately addressed all of my comments. The data newly added strengthened this manuscript significantly. Now I support this revision for publication after addressing the following minor comments.

Fig. 2e: Statistical tests are missing.

Re: We apologize for this oversight. Statistical tests have been added.

Fig. 3d: It would be helpful to outline the shape of the cells of interests (highlighted by arrows) by dotted line.

Re: The shape of the extruding cells has been outlined in the new version of the figure.

Reviewer #3 (Remarks to the Author):

The revised version of the manuscript by Gracia and colleagues is much improved and has addressed my comments, and as far as I could tell those made by the other two reviewers. In particular, the new experiments reported in Fig 5 establish that the actin cables are under tension and those in Fig 6 support the notion that those cables are essential for mesoderm invagination. I was also convinced by the new data reported in Figs 1&2. So altogether I reiterate my very strong support for publication of this work in Nature Communications.

I have a few minor comments:

1. Line 84: we could observe (no d)
2. Line 135: disappear (two p)
3. Line 147: add “novel” before knock-in. By the way, I apologize for not properly proofreading my original review in which I had written “Their observation of an apico-basal Myo2 cable, which had not been missed so far by all previous investigators is the key observation,” when I meant “which had been missed” and I hope it was clear to everyone.

Re: These modifications have been made.

4. I find that the experiment in which they “isolate the central region” not properly described even in the Methods section. I wasn’t entirely clear to me whether the authors prevented apical pulses in the anterior and posterior regions, fried the prospective mesoderm cells in those regions, or glued them to the yolk as previously done by Collinet and Lecuit. Please clarify.

Re: We clarify the effect of central domain isolation by apical cuts in the Methods section:

“central region was physically separated from the anterior and posterior region by the cuts but otherwise unperturbed. No cauterization occurs in these conditions, so each territory can move freely.”

5. Line 286: the word “alternative” is both badly chosen to describe what they mean and perhaps tainted these days (alternative facts...). I suggest that you write “the need for additional forces”.

Re: This has been changed.

6. Figure 1 & 2 legends panel a: the legend says small clones of cells expressing Snail in “fixed imaginal discs” or “generated in fixed imaginal discs”. Taken at face value, it looks contradictory to generate a clone in a fixed sample. So, I imagine that the authors mean that they are showing images taken from fixed samples in which they had generated clones prior to fixation. Please clarify.

Re: This was indeed misleading and has been clarified in the figure legend as follow:

“Snail-expressing clones (marked by GFP) generated in leg imaginal discs. Images of fixed tissues (top: apical view, bottom: sagittal view), showing the successive steps of cell apical constriction and delamination.”

7. A somewhat general comment about actin cables. Taking together Figs. 4a, 4a', 4c, 6c, it wasn't clear to me what is the distribution of actin cables in the prospective mesoderm as it undergoes invagination/EMT. Could you please quantify as to what is the distribution (1/cell, 1 every few cell). If the distribution is significantly below 1/cell, is the model described in Fig 8 still compatible and considering that possibility?

Re: We quantified the cable during the first half of the invagination process (the time frame during which apical surface of the cells can be follow accurately). These data are mentioned in the main text and described in Methods as follow:

Main text:

“Quantification of these structures reveals that they form in about half mesodermal cells during the first half of the invagination process (see Methods), strongly suggesting that each mesodermal cell form this structure at a given point during their delamination process.”

Methods:

“Quantification of cable-like structures of Myosin II

Using a 2.5 μ m Z projection of the most apical part of the ventral side of an sqhKI[eGFP], Ph-mcherry embryo, we quantified the appearance of new cable-like structure in mesodermal cells for each time point of the first half of the invagination process (the time frame during which apical surface of the cells can be follow accurately).”

8. In the modeling section of the Methods, add a few lines to contrast this new model to the many models that have been made to account for mesoderm invagination.

Re: We added a few lines explaining the novelty of this model compared to other models of mesoderm invagination:

“Comparison with other models: Mesoderm invagination has been extensively modeled. Most of the models proposed have been in 2D, representing a section of the embryo. The few 3D models developed so far were continuous models. The vertex model developed here has the advantage to take into account the specific dynamic of individual cells such as change in the number of cell neighbors (which is a strong limitation in continuous models) and thus is well adapted to mimic the heterogeneity in the timing of apical constriction and delamination that we observed. “

9. Please provide a video corresponding to 5a.

Re: A video has been added (Supplementary Movie 7).

We would like to thank the three reviewers for their constructive comments all along the reviewing process.